# Regularized linear autoencoders recover the principal components, eventually

**Xuchan Bao, James Lucas, Sushant Sachdeva, Roger Grosse**
University of Toronto;  Vector Institute
{jennybao,jlucas,sachdeva,rgrosse}@cs.toronto.edu

## Abstract

Our understanding of learning input-output relationships with neural nets has improved rapidly in recent years, but little is known about the convergence of the underlying representations, even in the simple case of linear autoencoders (LAEs). We show that when trained with proper regularization, LAEs can directly learn the optimal representation – ordered, axis-aligned principal components. We analyze two such regularization schemes: non-uniform $\ell_2$ regularization and a deterministic variant of nested dropout [24]. Though both regularization schemes converge to the optimal representation, we show that this convergence is slow due to ill-conditioning that worsens with increasing latent dimension. We show that the inefficiency of learning the optimal representation is not inevitable – we present a simple modification to the gradient descent update that greatly speeds up convergence empirically.[1]

## 1 Introduction

While there has been rapid progress in understanding the learning dynamics of neural networks, most such work focuses on the networks' ability to fit input-output relationships. However, many machine learning problems require learning representations with general utility. For example, the representations of a pre-trained neural network that successfully classifies the ImageNet dataset [6] may be reused for other tasks. It is difficult in general to analyze the dynamics of learning representations, as metrics such as training and validation accuracy reveal little about them. Furthermore, analysis through the Neural Tangent Kernel shows that in some settings, neural networks can learn input-output mappings without finding meaningful representations [11].

In some special cases, the optimal representations are known, allowing us to analyze representation learning exactly. In this paper, we focus on linear autoencoders (LAE). With specially chosen regularizers or update rules, their optimal weight representations consist of ordered, axis-aligned principal directions of the input data.

It is well known that the unregularized LAE finds solutions in the principal component spanning subspace [3], but in general, the individual components and corresponding eigenvalues cannot be recovered. This is because any invertible linear transformation and its inverse can be inserted between the encoder and the decoder without changing the loss. Kunin et al. [17] showed that applying $\ell_2$ regularization on the encoder and decoder reduces the symmetry of the stationary point solutions to the group of orthogonal transformations. The individual principal directions can then be recovered by applying the singular value decomposition (SVD) to the learned decoder weights.

We investigate how, with appropriate regularization, gradient-based optimization can further break the symmetry, and *directly* learn the individual principal directions. We analyze two such regularization schemes: *non-uniform* $\ell_2$ regularization and a deterministic variant of nested dropout [24].

The first regularization scheme we analyze applies *non-uniform* $\ell_2$ regularization on the weights connected to different latent dimensions. We show that at any global minimum, an LAE with non-uniform $\ell_2$ regularization directly recovers the ordered, axis-aligned principal components. We analyze the loss landscape and show that all local minima are global minima. The second scheme is nested dropout [24], which is already known to recover the individual principal components in the linear case.

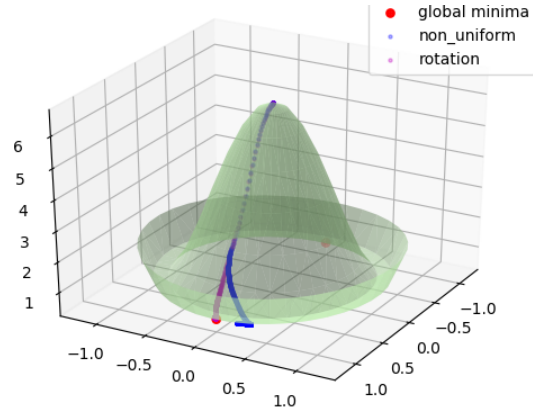

Figure 1: Visualization of the loss surface of an LAE with non-uniform $\ell_2$ regularization, plotted for a 2D subspace that includes a global optimal solution. The narrow valley along the rotation direction causes slow convergence. Detailed discussion can be found in Section 4.3.

After establishing two viable models, we ask: how fast can a gradient-based optimizer, such as gradient descent, find the correct representation? In principle, this ought to be a simple task once the PCA subspace is found, as an SVD on this low dimensional latent space can recover the correct alignment of the principal directions. However, we find that gradient descent applied to either aforementioned regularization scheme converges very slowly to the correct representation, even though the reconstruction error quickly decreases. To understand this phenomenon, we analyze the curvature of both objectives at their respective global minima, and show that these objectives cause ill-conditioning that worsens as the latent dimension is increased. Furthermore, we note that this ill-conditioning is nearly invisible in the training or validation loss, analogous to the general difficulty of measuring representation learning for practical nonlinear neural networks. The ill-conditioned loss landscape for non-uniform $\ell_2$ regularization is illustrated in Figure 1.

While the above results might suggest that gradient-based optimization is ill-suited for efficiently recovering the principal components, we show that this is not the case. We propose a simple iterative learning rule that recovers the principal components much faster than the previous methods. The gradient is augmented with a term that explicitly accounts for "rotation" of the latent space, and thus achieves a much stronger notion of symmetry breaking than the regularized objectives.

Our main contributions are as follows. 1) We characterize all stationary points of the non-uniform $\ell_2$ regularized objective, and prove it recovers the optimal representation at global minima (Section 4.1, 4.2). 2) Through analysis of Hessian conditioning, we explain the slow convergence of the non-uniform $\ell_2$ regularized LAE to the optimal representation (Section 4.3). 3) We derive a deterministic variant of nested dropout and explain its slow convergence with similar Hessian conditioning analysis (Section 5). 4) We propose an update rule that directly accounts for latent space rotation (Section 6). We prove that the gradient augmentation term globally drives the representation to be axis-aligned, and the update rule has local linear convergence to the optimal representation. We empirically show that this update rule accelerates learning the optimal representation.

## 2 Preliminaries

We consider linear models consisting of two weight matrices: an encoder $W_1 \in \mathbb{R}^{k \times m}$ and decoder $W_2 \in \mathbb{R}^{m \times k}$ (with $k < m$). The model learns a low-dimensional embedding of the data $X \in \mathbb{R}^{m \times n}$ (which we assume is zero-centered without loss of generality) by minimizing the objective,

$$\mathcal{L}(W_1, W_2; X) = \frac{1}{n}||X - W_2 W_1 X||_F^2 \tag{1}$$

We will assume $\sigma_1^2 > \cdots > \sigma_k^2 > 0$ are the $k$ largest eigenvalues of $\frac{1}{n}XX^\top$. The assumption that the $\sigma_1, \ldots, \sigma_k$ are positive and distinct ensures identifiability of the principal components, and is common in this setting [17]. Let $S = \text{diag}(\sigma_1, \ldots, \sigma_k)$. The corresponding eigenvectors are the columns of $U \in \mathbb{R}^{m \times k}$. Principal Component Analysis (PCA) [22] provides a unique optimal solution to this problem that can be interpreted as the projection of the data along columns of $U$, up to sign changes to the projection directions. However, the minima of (1) are not unique in general [17]. In fact, the objective is invariant under the transformation $(W_1, W_2) \mapsto (AW_1, W_2 A^{-1})$, for any invertible matrix $A \in \mathbb{R}^{k \times k}$.

**Regularized linear autoencoders.**   Kunin et al. [17] provide a theoretical analysis of $\ell_2$-regularized linear autoencoders, where the objective is as follows,

$$\mathcal{L}_\lambda(W_1, W_2; X) = \mathcal{L}(W_1, W_2; X) + \lambda\|W_1\|_F^2 + \lambda\|W_2\|_F^2. \tag{2}$$

Kunin et al. [17] proved that the set of globally optimal solutions to objective 2 exhibit only an orthogonal symmetry through the mapping: $(W_1, W_2) \mapsto (OW_1, W_2O^\top)$, for orthogonal matrix $O$.

## 3  Related work

Previous work has studied the exact recovery of the principal components in settings similar to LAEs. Rippel et al. [24] show that exact PCA can be recovered with an LAE by applying nested dropout on the hidden units. Nested dropout forces ordered information content in the hidden units. We derive and analyze a deterministic variant of nested dropout in Section 5. Connections between VAEs and probabilistic PCA (pPCA) have been explored before [19, 25, 5]. In particular, Lucas et al. [19] showed that a linear variational autoencoder (VAE) [15] with diagonal latent covariance trained with the evidence lower bound (ELBO) can learn the axis-aligned pPCA solution [29]. While this paper focuses on linear autoencoders and the full batch PCA problem, there exists an interesting connection between the proposed non-uniform $\ell_2$ regularization and the approach of Lucas et al. [19], as discussed in Appendix F. This connection was recently independently pointed out in the work of Kumar and Poole [16], who analyzed the implicit regularization effect of $\beta$-VAEs [10].

Kunin et al. [17] show that an LAE with uniform $\ell_2$ regularization reduces the symmetry group from $\mathrm{GL}_k(\mathbb{R})$ to $\mathrm{O}_k(\mathbb{R})$. They prove that the critical points of the $\ell_2$-regularized LAE are symmetric, and characterize the loss landscape. We adapt their insights to derive the loss landscape of LAEs with non-uniform $\ell_2$ regularization, and to prove identifiability at global optima. Concurrent work [20] addresses the identifiability issue in linear autoencoders by proposing a new loss function, that is a special case of deterministic nested dropout (discussed in Section 5), with a uniform prior distribution. Oftadeh et al. [20] show that the local minima correspond to ordered, axis-aligned representations. We show this in the general case, and additionally analyze the speed of convergence of this objective.

The rotation augmented gradient (RAG) proposed in Section 6 has connections to several existing algorithms. First, it is closely related to the Generalized Hebbian Algorithm (GHA) [26], which combines Oja's rule [21] with the Gram-Schmidt process. The detailed connection is discussed in Section 6.1. The GHA can also be used to derive a decentralized algorithm, as proposed in concurrent work [8], which casts PCA as a competitive game. Also, the RAG update appears to be in a similar form as the gradient of the Brockett cost function [1] on the Stiefel manifold, as discussed in Appendix G. However, the RAG update cannot be derived as the gradient of any loss function. Our proposed RAG update bears resemblance to the gradient masking approach in Spectral Inference Networks (SpIN) [23], which aims to learn ordered eigenfunctions of linear operators. The primary motivation of SpIN is to scale to learning eigenfunctions in extremely high-dimensional vector spaces. This is achieved by optimizing the Rayleigh quotient and achieving symmetry breaking through a novel application of the Cholesky decomposition to mask the gradient. This leads to a biased gradient that is corrected through the introduction of a bi-level optimization formulation for learning SpIN. RAG is not designed to learn arbitrary eigenfunctions but is able to achieve symmetry breaking without additional decomposition or bilevel optimization.

In this work, we discuss the weak symmetry breaking of regularized LAEs. Bamler and Mandt [4] address a similar problem for learning representations of time series data, which has weak symmetry in time. Through analysis of the Hessian matrix, they propose a new optimization algorithm – Goldstone gradient descent (Goldstone-GD) – that significantly speeds up convergence towards the correct alignment. The Goldstone-GD has interesting connection to the proposed RAG update in Section 6. RAG is analogous to applying the first order approximation of latent space rotation as an artificial gauge field, simultaneously with the full parameter update. We believe this is an exciting direction for future research.

Saxe et al. [27] study the continuous-time learning dynamics of linear autoencoders, and characterize the solutions under strict initialization conditions. Gidel et al. [9] extend this work along several important axes; they characterize the discrete-time dynamics for two-layer linear networks under relaxed (though still restricted) initialization conditions. Both Gidel et al. [9] and Arora et al. [2] also recognized a regularization effect of gradient descent, which encourages minimum norm solutions — the latter of which provides analysis for depth greater than two. These works provide exciting insight

into the capability of gradient-based optimization to learn meaningful representations, even when the loss function does not explicitly require such a representation. However, these works assume the covariance matrices of the input data and the latent code are co-diagonalizable, and do not analyze the dynamics of recovering rotation in the latent space. In contrast, in this work we study how effectively gradient descent is able to recover representations (including rotation in the latent space) in linear auto-encoders that are optimal for a designated objective.

# 4 Non-uniform $\ell_2$ weight regularization

In this section, we analyze linear autoencoders with non-uniform $\ell_2$ regularization where the rows and columns of $W_1$ and $W_2$ (respectively) are penalized with different weights. Let $0 < \lambda_1 < \cdots < \lambda_k$ be the $\ell_2$ penalty weights, and $\Lambda = \text{diag}(\lambda_1, \ldots, \lambda_k)$. The objective has the following form,

$$\mathcal{L}_{\sigma'}(W_1, W_2; X) = \frac{1}{n}||X - W_2 W_1 X||_F^2 + ||\Lambda^{1/2} W_1||_F^2 + ||W_2 \Lambda^{1/2}||_F^2 \tag{3}$$

We prove that the objective (3) has an ordered, axis-aligned global optimum, which can be learned using gradient based optimization. Intuitively, by penalizing different latent dimensions unequally, we force the LAE to explain higher variance directions with less heavily penalized latent dimensions.

The rest of this section proceeds as follows. First, we analyze the loss landscape of the objective (3) in section 4.1. Using this analysis, we show in section 4.2 that the global minimum recovers the ordered, axis-aligned individual principal directions. Moreover, all local minima are global minima. Section 4.3 explains mathematically the slow convergence to the optimal representation, by showing that at global optima, the Hessian of objective (3) is ill-conditioned.

## 4.1 Loss landscape

The analysis of the loss landscape is reminiscent of Kunin et al. [17]. We first prove the Transpose Theorem (Theorem 1) for objective (3). Then, we prove the Landscape Theorem (Theorem 2), which provides the analytical form of all stationary points of (3).

**Theorem 1.** *(Transpose Theorem) All stationary points of the objective* (3) *satisfy* $W_1 = W_2^\top$.

The proof is similar to that of Kunin et al. [17, Theorem 2.1.], and is deferred to Appendix E.1.

Theorem 1 enables us to proceed with a thorough analysis of the loss landscape of the non-uniform $\ell_2$ regularized LAE model. We fully characterize the stationary points of (3) in the following theorem.

**Theorem 2** (Landscape Theorem). *Assume* $\lambda_k < \sigma_k^2$. *All stationary points of* (3) *have the form:*

$$W_1^* = P(I - \Lambda S^{-2})^{\frac{1}{2}} U^\top \tag{4}$$

$$W_2^* = U(I - \Lambda S^{-2})^{\frac{1}{2}} P^\top \tag{5}$$

*where* $\mathcal{I} \subset \{1, \cdots, m\}$ *is an index set containing the indices of the learned components, and* $P \in \mathbb{R}^{k \times k}$ *has exactly one* $\pm 1$ *in each row and each column whose index is in* $\mathcal{I}$ *and zeros elsewhere.*

The full proof is deferred to Appendix E.2. Here we give intuition on this theorem and a proof sketch.

The uniform regularized objective in Kunin et al. [17] has orthogonal symmetry that is broken by the non-uniform $\ell_2$ regularization. In Theorem 2 we prove that the only remaining symmetries are (potentially reduced rank) permutations and reflections of the optimal representation. In fact, we will show in section 4.2 that at global minima, only reflection remains in the symmetry group.

*Proof of Theorem 2 (Sketch).* We consider applying a rotation matrix $R_{ij}$ and its inverse to $W_1$ and $W_2$ in the Landscape Theorem in Kunin et al. [17], respectively. $R_{ij}$ applies a rotation with angle $\theta$ on the plane spanned by the $i^{th}$ and $j^{th}$ latent dimensions. Under this rotation, the objective (3) is a cosine function with respect to $\theta$. In order for $\theta = 0$ to be a stationary point, the cosine function must have either amplitude 0 or phase $\beta\pi$ ($\beta \in \mathbb{Z}$). Finally, we prove that in the potentially reduced rank latent space, the symmetries are reduced to only permutations and reflections. $\square$

## 4.2 Recovery of ordered principal directions at global minima

Following the loss landscape analysis, we prove that the global minima of (3) correspond to ordered individual principal directions in the weights. Also, all local minima of (3) are global minima.

**Theorem 3.** *Assume $\lambda_k < \sigma_k^2$. The minimum value of* (3) *is achieved if and only if $W_1$ and $W_2$ are equal to* (4) *and* (5)*, with full rank and diagonal $P$. Moreover, all local minima are global minima.*

$P$ being full rank and diagonal corresponds to the columns of $W_2$ (and rows of $W_1$) being ordered, axis-aligned principal directions. The full proof is shown in Appendix E.3. Below is a sketch.

*Proof (Sketch).* Extending the proof for Theorem 2, in order for $\theta = 0$ to be a local minimum, we first show that $P$ must be full rank. Then, we show that the rows of $W_1^*$ (and columns of $W_2^*$) must be sorted in strictly descending order of magnitude, hence $P$ must be diagonal. It is then straightforward to show that the global optima are achieved if and only if $P$ is diagonal and full rank, and they correspond to ordered $k$ principal directions in the rows of the encoder (and columns of the decoder). Finally, we show that there does not exist a local minimum that is not global minimum. $\qquad\square$

### 4.3 Slow convergence to global minima

Theorem 3 ensures that a (perturbed) gradient based optimizer that efficiently escapes saddle points will eventually converge to a global optimum [7, 12]. However, we show in this section that this convergence is slow, due to ill-conditioning at global optima.

To gain better intuition about the loss landscape, consider Figure 1. The loss is plotted for a 2D subspace that includes a globally optimal solution of $W_1$ and $W_2$. More precisely, we use the parameterization $W_1 = \alpha O(I - \Lambda S^{-2})^{\frac{1}{2}} U^\top$, and $W_2 = W_1^\top$, where $\alpha$ is a scalar, and $O$ is a $2 \times 2$ rotation matrix parameterized by angle $\theta$. The $xy$-coordinate is obtained by $(\alpha \cos \theta, \alpha \sin \theta)$.

In general, narrow valleys in the loss landscape cause slow convergence. In the figure, we optimize $W_1$ and $W_2$ on this 2D subspace. We observe two distinct stages of the learning dynamics. The first stage is fast convergence to the correct subspace – the approximately circular "ring" of radius 1 with low loss. The fast convergence results from the steep slope along the radial direction. After converging to the subspace, there comes the very slow second stage of finding the optimal rotation angle — by moving through the narrow nearly-circular valley. This means that the symmetry breaking caused by the non-uniform $\ell_2$ regularization is a weak one.

We now formalize this intuition for general dimensions. The slow convergence to axis-aligned solutions is confirmed experimentally in full linear autoencoders in Section 7.

#### 4.3.1 Explaining slow convergence of the rotation

Denote the Hessian of objective (3) by $H$, and the largest and smallest eigenvalues of $H$ by $s_{\max}$ and $s_{\min}$ respectively. At a local minimum, the condition number $s_{\max}(H)/s_{\min}(H)$ determines the local convergence rate of gradient descent. Intuitively, the condition number characterizes the existence of narrow valleys in the loss landscape. Thus, we analyze the conditioning of the Hessian to better understand the slow convergence under non-uniform regularization.

In order to demonstrate ill-conditioning, we will lower bound the condition number through a lower bound on the largest eigenvalue, and an upper bound on the smallest. This is achieved by finding two vectors and computing the Rayleigh quotient, $f_H(v) = v^\top H v / v^\top v$ for each of them. Any Rayleigh quotient value is an upper (lower) bound on the smallest (largest) eigenvalue of $H$.

Looking back to Figure 1, we notice that the high-curvature direction is radial and corresponds to rescaling of the learned components while the low-curvature direction corresponds to rotation of the component axes. We compute the above Rayleigh quotient along these directions and combine to lower bound the overall condition number. The detailed derivation can be found in Appendix B. Ultimately, we show that the condition number can be lower bounded by,

$$\frac{2(k-1)(\sigma_1^2 - \sigma_k^2) \sum_{i=2}^{k-1}(\sigma_i^2 - \sigma_k^2)}{\sigma_1^2 \sigma_k^2}.$$

Depending on the distribution of the $\sigma$ values, as $k$ grows, the condition number quickly worsens. [2] This effect is observed empirically in Figure 4.

# 5 Deterministic nested dropout

The second regularization scheme we study is a deterministic variant of nested dropout [24]. Nested dropout is a stochastic algorithm for learning ordered representations in neural networks. In an LAE with $k$ hidden units, a prior distribution $p_B(\cdot)$ is assigned over the indices $1, \ldots, k$. When applying nested dropout, first an index $b \sim p_B(\cdot)$ is sampled, then all hidden units with indices $b + 1, \ldots, k$ are dropped. By imposing this dependency in the hidden unit dropout mask, nested dropout enforces an ordering of importance in the hidden units. Rippel et al. [24] proved that the global optimum of the nested dropout algorithm corresponds to the ordered, axis-aligned representation.

We propose a deterministic variant to the original nested dropout algorithm on LAEs, by replacing the stochastic loss function with its expectation. Taking the expectation eliminates the variance caused by stochasticity (which prevents the original nested dropout algorithm from converging to the exact PCA subspace), thereby making it directly comparable with other symmetry breaking techniques. Define $\pi_b$ as the operation setting hidden units with indices $b + 1, \ldots, k$ to zero. We define the loss here, and derive the analytical form in Appendix C.

$$\mathcal{L}_{\text{ND}}(W_1, W_2; X) = \mathbb{E}_{b \sim p_B(\cdot)}\Big[\frac{1}{2n}||X - W_2\pi_b(W_1 X)||_F^2\Big] \tag{6}$$

To find out how fast objective (6) is optimized with gradient-based optimizer, we adopt similar techniques as in Section 4.3 to analyze the condition number of the Hessian at the global optima. Derivation details are shown in Appendix D. The condition number is lower bounded by $\frac{8\sigma_1^2(k-1)^2}{\sigma_1^2 - \sigma_k^2}$.

Note that the lower bound assumes that the prior distribution $p_B(\cdot)$ is picked optimally with knowledge of $\sigma_1, \ldots, \sigma_k$. However, in practice we do not have access to $\sigma_1, \ldots, \sigma_k$ a priori, so the lower bound is loose. Nevertheless, the condition number grows at least quadratically in the latent dimension $k$. While the deterministic nested dropout might find the optimal representation efficiently in low dimensions, it fails to do so when $k$ is large. We confirm this observation empirically in Section 7.

# 6 Rotation augmented gradient for stronger symmetry breaking

The above analysis of regularized objectives may suggest that learning the correct representation in LAEs is inherently difficult for gradient-descent-like update rules. We now show that this is not the case by exhibiting a simple modification to the update rule which recovers the rotation efficiently. In particular, since learning the rotation of the latent space tends to be slow for non-uniform $\ell_2$ regularized LAE, we propose the rotation augmented gradient (RAG), which explicitly accounts for rotation in the latent space, as an alternative and more efficient method of symmetry breaking.

The RAG update is shown in Algorithm 1. Intuitively, RAG applies a simultaneous rotation on $W_1$ and $W_2$, aside from the usual gradient descent update of objective (1). To see this, notice that $A_t$ is skew-symmetric, so its matrix exponential is a rotation matrix. By Taylor expansion, we can see that RAG applies a first-order Taylor approximation of a rotation on $W_1$ and $W_2$.

$$\exp(\frac{\alpha}{n}A_t) = I + \sum_{i=1}^{\infty}\frac{\alpha^i}{i!n^i}A_t^i$$

The rest of this section aims to provide additional insight into RAG. Section 6.1 makes the connection to the Generalized Hebbian Algorithm (GHA) [26], a multi-dimensional variant of Oja's rule with global convergence. Section 6.2 points out an important property that greatly contributes to the stability of the algorithm: the rotation term in RAG conserves the reconstruction loss. Using this insight, Section 6.3 shows that the rotation term globally drives the solution to be axis-aligned. Finally, section 6.4 proves that RAG has local linear convergence to global minima.

## 6.1 Connection to the Generalized Hebbian Algorithm

RAG is closely related to the Generalized Hebbian Algorithm (GHA) [26]. To see the connection, we assume $W_1 = W_2^\top = W$.[3] For convenience, we drop the index $t$ in Algorithm 1. As in Algorithm 1,

**Algorithm 1** Rotation augmented gradient (RAG)

---

Given learning rate $\alpha_1$
Initialize $(W_1)_0, (W_2)_0$
**for** $t = 0 \ldots T - 1$ **do**
    $\nabla(W_1)_t = \nabla_{W_1}\mathcal{L}((W_1)_t, (W_2)_t)$
    $\nabla(W_2)_t = \nabla_{W_2}\mathcal{L}((W_1)_t, (W_2)_t)$

    $Y_t = (W_1)_t X$
    $A_t = \frac{1}{2}(\triangledown(Y_t Y_t^\top) - \triangle(Y_t Y_t^\top))$
    *($\triangledown$ (or $\triangle$) masks the lower (or upper) triangular part of a matrix (excluding the diagonal) with 0.)*

    $(W_1)_{t+1} \leftarrow (I + \frac{\alpha}{n}A_t)(W_1)_t - \alpha\nabla(W_1)_t$
    $(W_2)_{t+1} \leftarrow (W_2)_t(I - \frac{\alpha}{n}A_t) - \alpha\nabla(W_2)_t$
**end for**

---

$\triangle$ denotes the operation that masks the upper triangular part of a matrix (excluding the diagonal) with 0. With simple algebraic manipulation, the GHA and the RAG updates are compared below.

$$\mathbf{GHA}: W \leftarrow W + \frac{\alpha}{n}(YX^\top - \triangle(YY^\top)W)$$

$$\mathbf{RAG}: W \leftarrow W + \frac{\alpha}{n}[(YX^\top - \triangle(YY^\top)W) - \frac{1}{2}(YY^\top - \mathrm{diag}(YY^\top))W]$$

Compared to the GHA update, RAG has an additional term which, intuitively, decays certain notion of "correlation" between the columns in $W$ to zero. This additional term is important. As we will see in Section 6.2, the "non-reconstruction gradient term" of RAG conserves the reconstruction loss. This is a property that contributes to the training stability and that the GHA does not possess.

## 6.2 Rotation augmentation term conserves the reconstruction loss

An important property of RAG is that the addition of the rotation augmentation term conserves the reconstruction loss. To see this, we compare the instantaneous update for RAG and plain gradient descent on the unregularized objective (1).

We drop the index $t$ when analyzing the instantaneous update. We use superscripts $RAG$ and $GD$ to denote the instantaneous update following RAG and gradient descent on (1) respectively. We have,

$$\dot{W}_1^{RAG} = \dot{W}_1^{GD} + \frac{1}{n}AW_1, \quad \dot{W}_2^{RAG} = \dot{W}_2^{GD} - \frac{1}{n}W_2A.$$

Therefore, the rotation term conserves the reconstruction loss:

$$\frac{d}{dt}(W_2W_1)^{RAG} = \dot{W}_2^{RAG}W_1 + W_2\dot{W}_1^{RAG} = \dot{W}_2^{GD}W_1 + W_2\dot{W}_1^{GD} = \frac{d}{dt}(W_2W_1)^{GD}$$

$$\frac{d}{dt}\mathcal{L}(W_1, W_2)^{RAG} = \frac{d}{dt}\mathcal{L}(W_1, W_2)^{GD}.$$

This means that in RAG, learning the rotation is separated from learning the PCA subspace. The former is achieved with the rotation term, and the latter with the reconstruction gradient term. This is a desired property that contributes to the training stability.

## 6.3 Convergence of latent space rotation to axis-aligned solutions

The insight in Section 6.2 enables us to consider the subspace convergence and the rotation separately. We now prove that on the orthogonal subspace, the rotation term drives the weights to be axis-aligned. For better readability, we state this result below as an intuitive, informal theorem. The formal theorem and its proof are presented in Appendix E.5.

**Theorem 4** ((Informal) Global convergence to axis-aligned solutions)**.** *Initialized on the orthogonal subspace $W_1 = W_2^\top = OU^\top$, the instantaneous limit of RAG globally converges to the set of axis-aligned solutions, and the set of ordered, axis-aligned solutions is asymptotically stable.*

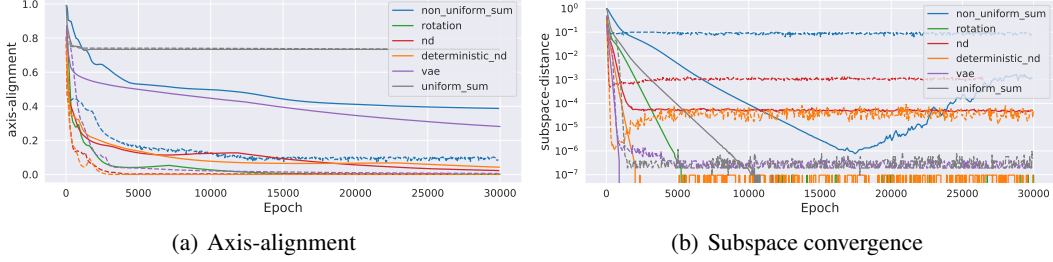

(a) Axis-alignment          (b) Subspace convergence

Figure 2: Learning dynamics of different LAE / linear VAE models trained on the MNIST ($k = 20$). Solid lines represent models trained using gradient descent with Nesterov momentum 0.9. Dashed lines represent models trained with Adam optimizer. The learning rate for each model and optimizer has been tuned to have the fastest convergence to axis-alignment.

### 6.4 Local linear convergence to the optimal representation

We show that RAG has local linear convergence to the ordered, axis-aligned solution. We show this in the limit of instantaneous update, and make the following assumptions.

It is reasonable to make Assumption 1, since learning the PCA subspace is usually much more efficient than the rotation. Also, Section 6.2 has shown that the rotational update term conserves the reconstruction loss, thus can be analyzed independently. Assumptions 2 and 3 state that we focus on the convergence *local* to the ordered, axis-aligned solution.

> **Assumption 1.** *The PCA subspace is recovered, i.e. the gradient due to reconstruction loss is 0.*
>
> **Assumption 2.** $YY^\top$ *is diagonally dominant with factor* $0 < \epsilon \ll 1$, *i.e.* $\sum_{j \neq i} |(YY^\top)_{ij}| < \epsilon \cdot (YY^\top)_{ii}$.
>
> **Assumption 3.** *The diagonal elements of* $YY^\top$ *are positive and sorted in strict descending order, i.e.* $\forall\, i < j,\ (YY^\top)_{ii} > (YY^\top)_{jj} > 0$.

**Definition 6.1.** The "non-diagonality" of a matrix $M \in \mathbb{R}^{k \times k}$ is $Nd(M) = \sum_{i=1}^{k} \sum_{j=1, j \neq i}^{k} |M_{ij}|$.

**Theorem 5** (Local Linear Convergence)**.** *Let* $g = \min_{i,j,i \neq j} \frac{1}{n} |(YY^\top)_{ii} - (YY^\top)_{jj}|$. *With Assumption 1-3 and in the instantaneous limit ($\alpha \to 0$), for an LAE updated with RAG, $Nd(\frac{1}{n} YY^\top)$ converges to 0 with an instantaneous linear rate of $g$.*

The proof is deferred to Appendix E.4. Note that the optimal representation corresponds to diagonal $\frac{1}{n} YY^\top$ with ordered diagonal elements, which RAG has local linear convergence to. Note that near the global optimum, $g$ is approximately the smallest "gap" between the eigenvalues of $\frac{1}{n} XX^\top$.

## 7 Experiments

In this section, we seek answers to these questions: 1) What is the empirical speed of convergence of an LAE to the ordered, axis-aligned solution using gradient-based optimization, with the aforementioned objectives or update rules? 2) How is the learning dynamics affected by different gradient-based optimizers? 3) How does the convergence speed scale to different sizes of the latent representations?

First, we define the metrics for axis-alignment and subspace convergence using the learned $W_2$ (Definitions 7.1 and 7.2). Definition 7.2 is equal to the Definition 1 in Tang [28] scaled by $\frac{1}{k}$.

**Definition 7.1** (Distance to axis-aligned solution)**.** We define the distance to the axis-aligned solution as $d_{\text{align}}(W_2, U) = 1 - \frac{1}{k} \sum_{i=1}^{k} \max_j \frac{(U_i^\top (W_2)_j)^2}{||U_i||_2^2 ||(W_2)_j||_2^2}$ (subscripts represent the column index).

**Definition 7.2** (Distance to optimal subspace)**.** Let $U_{W_2} \in \mathbb{R}^{m \times k}$ consist of the left singular vectors of $W_2$. We define the distance to the optimal subspace as $d_{\text{sub}}(W_2, U) = 1 - \frac{1}{k} \text{Tr}(UU^\top U_{W_2} U_{W_2}^\top)$.

**Convergence to optimal representation**    We compare the learning dynamics for six models: uniform and non-uniform $\ell_2$ regularized LAEs, LAE updated with the RAG, LAEs updated with nested dropout and its deterministic variant, and linear VAE with diagonal latent covariance [19].

Figure 2 and 3 show the learning dynamics of these model on the MNIST dataset [18], with $k = 20$. Further details can be found in Appendix H. We use full-batch training for this experiment, which is sufficient to demonstrate the symmetry breaking properties of these models. For completeness, we also show mini-batch experiments in Appendix I.2. Figure 2 shows the evolution of the two metrics:

distance to axis-alignment and to the optimal subspace, when the models are trained with Nesterov accelerated gradient descent and the Adam optimizer [14], respectively. Figure 3 visualizes the matrix $U^\top W_2$, and the first 20 learned principal components of MNIST (columns of $W_2$).

Unsurprisingly, the uniform $\ell_2$ regularization fails to learn the axis-aligned solutions. When optimized with Nesterov accelerated gradient descent, the regularized models, especially non-uniform $\ell_2$ regularization, has slow convergence to the axis-aligned solution. The model trained with RAG has a faster convergence. It's worth noting that Adam optimizer accelerates the learning of the regularized models and the linear VAE, but it is not directly applicable to RAG.

**Scalability to latent representation sizes**  As predicted by the Hessian condition number analysis in Section 4.3 and Section 5, we expect the models with the two regularized objectives to become much less efficient as the latent dimension grows. We test this on a synthetic dataset with input dimension $m = 1000$. The data singular values are $1, \dots, m$. Full experimental details are in Appendix H. Figure 4 shows how quickly each model converges to the axis-alignment distance of 0.3. When optimized with the Nesterov accelerated gradient descent, the non-uniform $\ell_2$ regularization and the deterministic nested dropout scale poorly with latent dimension compared to RAG. This result is consistent with our Hessian condition number analysis. Although Adam optimizer provides acceleration for the regularized objectives, it does not solve the poor scaling with latent dimensions, as both regularized models fail to converge with large latent dimensions.

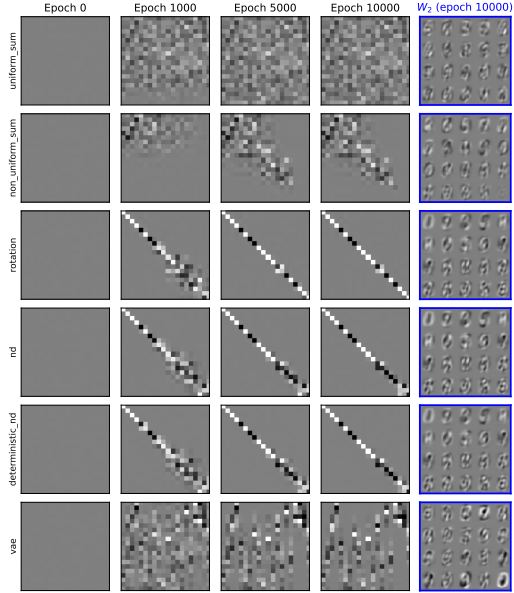

Figure 3: Visualization of $U^\top W_2$ and the decoder weights (last column) of LAEs trained on MNIST. All models are trained with Nesterov accelerated gradient descent. Pixel values range between -1 (black) and 1 (white). An ordered, axis-aligned solution corresponds to diagonal $U^\top W_2$ with $\pm 1$ diagonal entries. The linear VAE does not enforce order over the hidden dimensions, so $U^\top W_2$ will resemble a permutation matrix at convergence.

## 8   Conclusion

Learning the optimal representation in an LAE amounts to symmetry breaking, which is central to general representation learning. In this work, we investigated several algorithms that learn the optimal representation in LAEs, and analyze their strength of symmetry breaking. We showed that naive regularization approaches are able to break the symmetry in LAEs but introduce ill-conditioning that leads to slow convergence. The alternative algorithm we proposed, the rotation augmented gradient (RAG), guarantees convergence to the optimal representation and overcomes the convergence speed issues present in the regularization approaches. Our theoretical analysis provides new insights into the loss landscape of representation learning problems and the algorithmic properties required to perform gradient-based learning of representations.

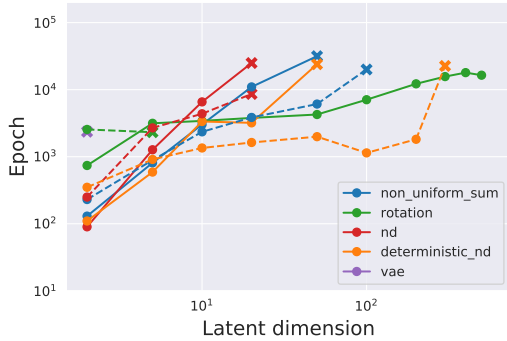

Figure 4: Epochs taken to reach 0.3 axis-alignment distance on the synthetic dataset, for different latent dimensions. Solid and dashed lines represent models trained with Nesterov accelerated gradient descent and Adam optimizer respectively. Cross markers indicate that beyond the current latent dimension, the models fail to reach 0.3 axis-alignment distance within 50k epochs.

## Broader Impact

The contribution of this work is the theoretical understanding of learning the optimal representations in LAEs with gradient-based optimizers. We believe that the discussion of broader impact is not applicable to this work.

## Acknowledgments and Disclosure of Funding

We thank Jonathan Bloom, Richard Zemel, Juhan Bae and Cem Anil for helpful discussions.

XB is supported by a Natural Sciences and Engineering Research council (NSERC) Discovery Grant. JL is supported by grants from NSERC and Samsung. SS's research is supported in part by an NSERC Discovery grant. RG acknowledges support from the CIFAR Canadian AI Chairs program. Part of this research was conducted when SS and RG were visitors at the Special year on Optimization, Statistics, and Theoretical Machine Learning at the School of Mathematics, Institute for Advanced Study, Princeton.

## Footnotes

[1]The code is available at `https://github.com/XuchanBao/linear-ae`

[2]Note that the lower bound is derived assuming the $\lambda$ values are optimally chosen, when $\sigma$ values are known. In practice, this is generally infeasible because 1) we do not have access to the $\sigma$ values, and 2) the $\lambda$ values that minimize the Hessian condition number at global minima may slow down the earlier phase of training, when the weights are far from the global optima, as shown experimentally in Appendix I.1. The difficulty of choosing an optimal set of $\lambda$ values contributes to the weakness of symmetry breaking by the non-uniform $\ell_2$ regularization.

[3]$W_1 = W_2^\top$ is required by the GHA. For RAG, this can be achieved by using balanced initialization $(W_1)_0 = (W_2)_0^\top$, as RAG stays balanced if initialized so.

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
