[Supplementary Material]

## A  Table of Notation

| | Description |
|---|---|
| $k$ | Number of latent dimensions in hidden layer of autoencoder |
| $m$ | Number of dimensions of input data |
| $n$ | Number of datapoints |
| $W_1 \in \mathbb{R}^{k \times m}$ | Encoder weight matrix |
| $W_2 \in \mathbb{R}^{m \times k}$ | Decoder weight matrix |
| $X \in \mathbb{R}^{m \times n}$ | Data matrix, with $n$ $m$-dimensional |
| $\|\cdot\|_F$ | The Frobenius matrix norm |
| $\sigma_i^2$ | The $i^{th}$ eigenvalues of the empirical covariance matrix $\frac{1}{n}XX^\top$ |
| $S$ | Diagonal matrix with entries $\sigma_1, \ldots, \sigma_k$ |
| $U$ | Matrix whose columns are the eigenvectors of $\frac{1}{n}XX^\top$, in descending order of corresponding eigenvalues |
| $\mathcal{L}$ | Linear autoencoder reconstruction loss function |
| $\mathcal{L}_\lambda$ | Linear autoencoder loss function with uniform $\ell_2$ regularization |
| $\mathcal{L}_{\sigma'}$ | Linear autoencoder loss function with uniform $\ell_2$ regularization |
| $\Lambda$ | Diagonal matrix containing non-uniform regularization weights, $\text{diag}(\lambda_1, \ldots, \lambda_k)$ |
| $H$ | The Hessian matrix of the non-uniform regularized loss (unless otherwise specified) |
| $s_{\max}(H)$ | The largest eigenvalue of $H$ |
| $s_{\min}(H)$ | The smallest eigenvalue of $H$ |
| $f_A(v)$ | The Rayleigh quotient, $f_A(v) = v^\top A v / v^\top v$ |
| $\mathcal{L}_{\text{ND}}$ | Linear autoencoder with nested dropout loss function |
| $Y$ | $Y = W_1 X$, latent representation of linear autoencoder |
| $\alpha$ | Learning rate of gradient descent optimizer |
| $\triangledown(\cdot) / \triangle(\cdot)$ | Operator that sets the lower or upper triangular part (excluding the diagonal) to zero of a matrix (respectively) |

Table 1: Summary of notation used in this manuscript, ordered according to introduction in main text.

## B  Conditioning analysis for the regularized LAE

Our goal here is to show that the regularized LAE objective is ill-conditioned, and also to provide insight into the nature of the ill-conditioning. In order to demonstrate ill-conditioning, we will prove a lower bound on the condition number of the Hessian at a minimum, by providing a lower bound on the largest singular value of the Hessian and an upper bound on the smallest singular value. The largest eigenvalue limits the maximum stable learning rate, and thus if the ratio of these two terms is very large then we will be forced to make slow progress in learning the correct rotation. Throughout this section, we will assume that the data covariance is full rank and has unique eigenvalues.

Since the Hessian $H$ is symmetric, we can compute bounds on the singular values through the Rayleigh quotient, $f_H(v) = v^\top H v / v^\top v$. In particular, for any vector $v$ of appropriate dimensions,

$$s_{\min}(H) \leq f_H(v) \leq s_{\max}(H). \tag{7}$$

Thus, if we exhibit two vectors with Rayleigh quotients $f_H(v_1)$ and $f_H(v_2)$, then the condition number is lower bounded by $f_H(v_1)/f_H(v_2)$.

In order to compute the Rayleigh quotient, we compute the second derivatives of auxiliary functions parameterizing the loss over paths in weight-space, about the globally optimal weights. This can be justified by the following Lemma,

**Lemma 1.** *Consider smooth functions $\ell : \mathbb{R}^n \to \mathbb{R}$, and $g : \mathbb{R} \to \mathbb{R}^n$, with $h = \ell \circ g : \mathbb{R} \to \mathbb{R}$. Assume that $g(0)$ is a stationary point of $\ell$, and let $H$ denote the Hessian of $\ell$ at $g(0)$. Writing $f_H(v)$ for the Rayleigh quotient of $H$ with $v$, we have,*

$$f_H(v) = \frac{h''(0)}{J_g(0)^\top J_g(0)},$$

*where $J_g$ denotes the Jacobian of $g$.*

*Proof.* The proof is a simple application of the chain rule and Taylor's theorem. Let $u = g(\alpha)$, then,

$$\frac{d^2 h}{d\alpha^2} = J_g^\top \frac{\partial^2 \ell}{\partial^2 u} J_g + \frac{d\ell}{du}^\top \frac{d^2 g}{d\alpha^2}.$$

Thus, by Taylor expanding $h$ about $\alpha = 0$,

$$h(\alpha) = h(0) + \alpha \frac{dh}{d\alpha}\Big|_{\alpha=0} + \frac{\alpha^2}{2} \frac{d^2 h}{d\alpha^2}\Big|_{\alpha=0} + o(\alpha^3) \tag{8}$$

$$= h(0) + \alpha \left(\frac{d\ell}{du}^\top J_g\right)\Big|_{\alpha=0} + \frac{\alpha^2}{2}\left(J_g^\top \frac{\partial^2 \ell}{\partial^2 u} J_g + \frac{d\ell}{du}^\top \frac{d^2 g}{d\alpha^2}\right)\Big|_{\alpha=0} + o(\alpha^3) \tag{9}$$

Now, note that as $g(0)$ is a stationary point of $\ell$, thus $\frac{d\ell}{du}\big|_{\alpha=0} = 0$. Differentiating the Taylor expansion twice with respect to $\alpha$, and evaluating at $\alpha = 0$ gives,

$$h''(0) = J_g(0)^\top H J_g(0)$$

Thus, dividing by $J_g(0)^\top J_g(0)$ we recover the Rayleigh quotient at $H$. $\square$

**Scaling curvature** The first vector for which we compute the Rayleigh quotient corresponds to rescaling of the largest principal component at the global optimum. To do so, we define the auxiliary function,

$$h_Z(\alpha) = \mathcal{L}_{\sigma'}(W_1 + \alpha Z_1, W_2 + \alpha Z_2; X)$$
$$= \frac{1}{2n}\|X - (W_2 + \alpha Z_2)(W_1 + \alpha Z_1)X\|_F^2 + \frac{1}{2}\|\Lambda^{1/2}(W_1 + \alpha Z_1)\|_F^2 + \frac{1}{2}\|(W_2 + \alpha Z_2)\Lambda^{1/2}\|_F^2$$

Thus, by Lemma 1, we have $h_Z''(0) = \frac{1}{2}\text{vec}(\begin{bmatrix} Z_1^\top & Z_2 \end{bmatrix})^\top H \text{vec}(\begin{bmatrix} Z_1^\top & Z_2 \end{bmatrix})$, that is, the curvature evaluated along the direction $\begin{bmatrix} Z_1^\top & Z_2 \end{bmatrix}$. It is easy to see that $h_Z(\alpha)$ is a polynomial in $\alpha$, and thus to evaluate $h_Z''(0)$ we need only compute the terms of order $\alpha^2$ in $h_Z$. Writing the objective using the trace operation,

$$h_Z(\alpha) = \frac{1}{2n}\text{Tr}\big[(X - W_2 W_1 X - \alpha(Z_2 W_1 + W_2 Z_1)X - \alpha^2 Z_2 Z_1 X)^\top$$
$$(X - W_2 W_1 X - \alpha(Z_2 W_1 + W_2 Z_1)X - \alpha^2 Z_2 Z_1 X)\big]$$
$$+ \frac{1}{2}\text{Tr}\big[\Lambda((W_1 + \alpha Z_1)(W_1 + \alpha Z_1)^\top + (W_2 + \alpha Z_2)^\top(W_2 + \alpha Z_2))\big]$$

Collecting the terms in $\alpha^2$:

$$\alpha^2\big(\frac{1}{2n}\text{Tr}\big[X^\top(Z_2 W_1 + W_2 Z_1)^\top(Z_2 W_1 + W_2 Z_1)X - 2X^\top Z_1^\top Z_2^\top(X - W_2 W_1 X)\big]$$
$$+ \frac{1}{2}\text{Tr}\big[\Lambda(Z_1 Z_1^\top + Z_2^\top Z_2)\big]\big)$$

Above we have used permutation invariance of the trace operator to collect together two middle terms.

At this point, we proceed by analyzing the Rayleigh quotient along the direction corresponding to scaling the leading principal component column, at the global optimum:

$$W_1^\top = W_2 = W = U(I - \Lambda S^{-2})^{\frac{1}{2}}$$

where $U$ are the eigenvectors of the data covariance, and $S^2$ the diagonal matrix containing the corresponding eigenvalues. Additionally, we choose $Z_1$ and $Z_2$ to contain the first column of the decoder ($w_1$), padded with zeros to match the dimension of $W_1$ and $W_2$,

$$Z_1^\top = Z_2 = Z = (\ w_1 \quad 0\ ),$$

We will require the following identities,

$$X - W_2 W_1 X = nU(S - (I - \Lambda S^{-2})S)V^\top = nU\Lambda S^{-1}V^\top \tag{10}$$

$$U^\top W = (I - \Lambda S^{-2})^{\frac{1}{2}} \tag{11}$$

$$U^\top Z = \begin{pmatrix} \sqrt{1 - \lambda_1 \sigma_1^{-2}} & 0 \\ 0 & 0 \end{pmatrix} \tag{12}$$

$$W^\top W = I - \Lambda S^{-2} \tag{13}$$

$$Z^\top Z = \begin{pmatrix} 1 - \lambda_1 \sigma_1^{-2} & 0 \\ 0 & 0 \end{pmatrix} \tag{14}$$

$$Z^\top W = \begin{pmatrix} 1 - \lambda_1 \sigma_1^{-2} & 0 \\ 0 & 0 \end{pmatrix} \tag{15}$$

We now tackle each term in turn. Beginning with the first,

$$\begin{aligned}
&\mathrm{Tr}\left(X^\top(Z_2 W_1 + W_2 Z_1)^\top(Z_2 W_1 + W_2 Z_1)X\right) \\
&= \mathrm{Tr}\left(XX^\top(ZW^\top + WZ^\top)(ZW^\top + WZ^\top)\right) \\
&= n\mathrm{Tr}\left(S^2 U^\top(ZW^\top + WZ^\top)(ZW^\top + WZ^\top)U\right) \\
&= n\mathrm{Tr}\left(S^2(U^\top ZW^\top + U^\top WZ^\top)(Z(U^\top W)^\top + W(U^\top Z)^\top)\right) \\
&= n\mathrm{Tr}\big(S^2((U^\top Z)(W^\top Z)(U^\top W)^\top + (U^\top Z)(W^\top W)(U^\top Z)^\top \\
&\quad + (U^\top W)(Z^\top Z)(U^\top W)^\top + (U^\top W)(Z^\top W)(U^\top Z)^\top\big) \\
&= 4n\sigma_1^2(1 - \lambda_1 \sigma_1^{-2})^2
\end{aligned}$$

For the second term,

$$\begin{aligned}
-2\mathrm{Tr}\left(X^\top Z_1^\top Z_2^\top(X - W_2 W_1 X)\right) &= -2n\mathrm{Tr}\left(VSU^\top ZZ^\top U\Lambda S^{-1}V^\top\right) \\
&= -2n\mathrm{Tr}\left(U^\top ZZ^\top U\Lambda\right) \\
&= -2n\lambda_1(1 - \lambda_1 \sigma_1^{-2})
\end{aligned}$$

For the final third term,

$$\mathrm{Tr}\left(\Lambda(Z_1 Z_1^\top + Z_2^\top Z_2)\right) = 2\mathrm{Tr}\left(\Lambda(Z^\top Z)\right) = 2\lambda_1(1 - \lambda_1 \sigma_1^{-2})$$

Combining these,

$$h_Z''(0) = (1 - \lambda_1 \sigma_1^{-2})\left(4\sigma_1^2(1 - \lambda_1 \sigma_1^{-2}) + 2\lambda_1 - 2\lambda_1\right) = 4\sigma_1^2(1 - \lambda_1 \sigma_1^{-2})^2$$

Using Lemma 1, we see that to recover the Rayleigh quotient, we must divide by $\|\begin{bmatrix} Z_1^\top & Z_2 \end{bmatrix}\|_F^2 = 2(1 - \lambda_1 \sigma_1^{-2})$. Thus, using Equation 7, we have

$$s_{\max}(H) \geq \frac{\mathrm{vec}(\begin{bmatrix} Z_1^\top & Z_2 \end{bmatrix})^\top H \mathrm{vec}(\begin{bmatrix} Z_1^\top & Z_2 \end{bmatrix})}{\|\mathrm{vec}(\begin{bmatrix} Z_1^\top & Z_2 \end{bmatrix})\|_F^2} = 2\sigma_1^2(1 - \lambda_1 \sigma_1^{-2}) \geq 2(\sigma_1^2 - \sigma_k^2).$$

**Rotation curvature** To approximate the rotation curvature, we consider paths along the rotation manifold. This corresponds to rotating the latent space of the LAE. Using Lemma 1, we will compute the Rayleigh quotient $f_H(t)$ for vectors $t$ on the tangent space to this rotation manifold.

Explicitly, we consider an auxiliary function of the form,

$$\gamma_R(\theta) = \frac{1}{2n}\|X - W_2 R(\theta)^\top R(\theta) W_1 X\|_F^2 + \frac{1}{2}\|\Lambda^{1/2} R(\theta) W_1\|_F^2 + \frac{1}{2}\|W_2 R(\theta)^\top \Lambda^{1/2}\|_F^2,$$

where $R(\theta)$ is a rotation matrix parameterized by $\theta$. The first term does not depend on $\theta$, as $R$ is orthogonal. Thus, we need only compute the second derivative of the regularization terms. About the global optimum, the regularization terms can be written as,

$$\mathrm{Tr}\left(\Lambda R(\theta) W^T W R(\theta)^T\right)$$

We will consider rotations of the $i^{th}$ and $j^{th}$ columns only (a Givens rotation). To reduce notational clutter, we write $\nu_i = (1 - \lambda_i \sigma_i^{-2})$.

$$
\begin{aligned}
\mathrm{Tr}\left(\Lambda R(\theta) W^T W R(\theta)^T\right) &= \mathrm{Tr}\left(\Lambda \begin{bmatrix} \nu_i \cos\theta & -\nu_j \sin\theta \\ \nu_i \sin\theta & \nu_j \cos\theta \end{bmatrix} \begin{bmatrix} \cos\theta & \sin\theta \\ -\sin\theta & \cos\theta \end{bmatrix}\right) + \sum_{l \neq i,j} \lambda_l \nu_l \\
&= \mathrm{Tr}\left(\Lambda \begin{bmatrix} \nu_i \cos^2\theta + \nu_j \sin^2\theta & \cdot \\ \cdot & \nu_i \sin^2\theta + \nu_j \cos^2\theta \end{bmatrix}\right) + \sum_{l \neq i,j} \lambda_l \nu_l \\
&= \lambda_i(\nu_i \cos^2\theta + \nu_j \sin^2\theta) + \lambda_j(\nu_i \sin^2\theta + \nu_j \cos^2\theta) + \sum_{l \neq i,j} \lambda_l \nu_l \\
&= \nu_i(\lambda_i - \lambda_j)\cos^2\theta + \nu_j(\lambda_i - \lambda_j)\sin^2\theta + \sum_{l \neq i,j} \lambda_l \nu_l
\end{aligned}
$$

We proceed to take derivatives.

$$\frac{\partial}{\partial\theta}\mathrm{Tr}\left(\Lambda R(\theta) W^T W R(\theta)^T\right) = 2\sin\theta\cos\theta(\nu_j - \nu_i)(\lambda_i - \lambda_j) = \sin 2\theta(\nu_j - \nu_i)(\lambda_i - \lambda_j)$$

Thus, the second derivative, $\gamma''(\theta)$, is given by,

$$2(\nu_j - \nu_i)(\lambda_i - \lambda_j)\cos 2\theta$$

Which, when evaluated at $\theta = 0$, gives,

$$\gamma''(0) = 2(\nu_j - \nu_i)(\lambda_i - \lambda_j).$$

Per Lemma 1, we also require the magnitude of the tangent to the path at $\theta = 0$, to compute the Rayleigh quotient. At the global optimum, we have,

$$
\begin{aligned}
\left\|W\frac{d}{d\theta}R(\theta)^\top\right\|_F^2 &= \left\|(I - \Lambda S^{-2})^{1/2}\frac{d}{d\theta}R(\theta)^\top\right\|_F^2 \\
&= \left\|\begin{bmatrix} \nu_i^{1/2} & 0 \\ 0 & \nu_j^{1/2} \end{bmatrix}\begin{bmatrix} -\sin\theta & \cos\theta \\ -\cos\theta & -\sin\theta \end{bmatrix}\right\|_f^2 \\
&= \nu_i + \nu_j
\end{aligned}
$$

Thus the Rayleigh quotient is given by,

$$f_H(t) = \frac{\nu_j - \nu_i}{\nu_i + \nu_j}(\lambda_i - \lambda_j).$$

Without loss of generality, we will pick $i > j$, so that $\lambda_i > \lambda_j$, $\sigma_i < \sigma_j$, and $\nu_i < \nu_j$. Where the last of these inequalities follows from $\lambda_i \sigma_i^{-2} > \lambda_i \sigma_j^{-2} > \lambda_j \sigma_j^{-2}$.

**Conditioning of the objective** We can combine the lower bound on the largest singular value with the upper bound on the smallest singular value to give a lower bound on the condition number. The ratio can be written,

$$\frac{2(\sigma_1^2 - \sigma_k^2)(\nu_i + \nu_j)}{(\lambda_i - \lambda_j)(\nu_j - \nu_i)}$$

Thus, the condition number is controlled by our choice of placement of $\{\lambda_j\}_{j=1}^k$ on the interval $(0, \sigma_k^2)$. We lower bound the condition number by the solution to the following optimization problem,

$$\text{cond}(H_\Lambda) \geq \min_{\lambda_1,\ldots,\lambda_k} \max_{i>j} \frac{2(\sigma_1^2 - \sigma_k^2)(\nu_i + \nu_j)}{(\lambda_i - \lambda_j)(\nu_j - \nu_i)} \tag{16}$$

To simplify the problem, we lower bound $\nu_i + \nu_j > 2\nu_i$. Now the inner maximization can be reduced to a search over a single index by setting $i = j + 1$, as the entries of $\Lambda$ and each $\nu$ are monotonic (decreasing and increasing respectively).

Further, we can see that at the minimum each of the terms $\nu_{j+1}/\left((\lambda_{j+1} - \lambda_j)(\nu_j - \nu_{j+1})\right)$ must be equal — otherwise we could adjust our choice of $\Lambda$ to reduce the largest of these terms. We denote the equal value as $c_1$. Thus, we can write,

$$\lambda_k - \lambda_1 = \sum_{j=1}^{k-1}(\lambda_{j+1} - \lambda_j) = \frac{1}{c_1}\sum_{j=1}^{k-1}\frac{\nu_{j+1}}{\nu_j - \nu_{j+1}}$$

$$\implies c_1 = \frac{1}{\lambda_k - \lambda_1}\sum_{j=1}^{k-1}\frac{\nu_{j+1}}{\nu_j - \nu_{j+1}} > \frac{1}{\sigma_k^2}\sum_{j=1}^{k-1}\frac{\nu_{j+1}}{\nu_j - \nu_{j+1}} \tag{17}$$

We can further bound $c_1$ by finding a lower bound for the summation in (17). The minimum of (17) can be reached when all terms in the summation are equal. To see this, we let the value of each summation term to be $c_2 > 0$. We have,

$$\nu_{j+1} = \frac{c_2}{1 + c_2}\nu_j, \quad j = 1,\ldots,k-1$$

For $l = 2,\ldots,k-1$, the derivative of (17) with respect to $\nu_l$ is zero, and the second derivative is positive.

$$\frac{\partial}{\partial \nu_l}\frac{1}{\sigma_k^2}\sum_{j=1}^{k-1}\frac{\nu_{j+1}}{\nu_j - \nu_{j+1}} = \frac{1}{\sigma_k^2}\frac{\partial}{\partial \nu_l}\left(\frac{\nu_{l-1}}{\nu_{l-1} - \nu_l} + \frac{\nu_{l+1}}{\nu_l - \nu_{l+1}}\right)$$

$$= \frac{1}{\sigma_k^2}\left(\frac{\nu_{l-1}}{(\nu_{l-1} - \nu_l)^2} - \frac{\nu_{l+1}}{(\nu_l - \nu_{l+1})^2}\right)$$

$$= \frac{1}{\sigma_k^2}\cdot\frac{1}{\nu_l}\left(\frac{\frac{1+c_2}{c_2}}{(\frac{1+c_2}{c_2} - 1)^2} - \frac{\frac{c_2}{1+c_2}}{(1 - \frac{c_2}{1+c_2})^2}\right)$$

$$= 0$$

$$\frac{\partial^2}{\partial \nu_l^2}\frac{1}{\sigma_k^2}\sum_{j=1}^{k-1}\frac{\nu_{j+1}}{\nu_j - \nu_{j+1}} = \frac{1}{\sigma_k^2}\left(\frac{2\nu_{l-1}(\nu_{l-1} - \nu_l)}{(\nu_{l-1} - \nu_l)^4} + \frac{2\nu_{l+1}(\nu_l - \nu_{l+1})}{(\nu_l - \nu_{l+1})^4}\right) > 0$$

Therefore, the minimum of (17) can be reached when all terms in the summation are equal. We bound $c_2$ as follows,

$$\nu_1 - \nu_k = \sum_{j=1}^{k-1}(\nu_j - \nu_{j+1}) = \frac{1}{c_2}\sum_{j=1}^{k-1}\nu_{j+1}$$

$$\implies c_2 = \frac{1}{\nu_1 - \nu_k}\sum_{j=1}^{k-1}\nu_{j+1} > \frac{1}{\nu_1}\sum_{i=2}^{k}\left(1 - \frac{\lambda_i}{\sigma_i^2}\right) > \sum_{i=2}^{k}\frac{\sigma_i^2 - \lambda_i}{\sigma_i^2} > \frac{1}{\sigma_1^2}\sum_{i=2}^{k-1}(\sigma_i^2 - \sigma_k^2)$$

We bound the condition number by putting the above step together,

$$\text{cond}(H_\Lambda) \geq 2(\sigma_1^2 - \sigma_k^2)c_1 > 2(\sigma_1^2 - \sigma_k^2)\frac{k-1}{\sigma_k^2}c_2 > \frac{2(k-1)(\sigma_1^2 - \sigma_k^2)\sum_{i=2}^{k-1}(\sigma_i^2 - \sigma_k^2)}{\sigma_1^2\sigma_k^2}$$

# C   Deterministic nested dropout derivation

In this section, we derive the analytical form of the expected LAE loss of the nested dropout algorithm [24].

As in Section 5, we define $\pi_b$ as the operation that sets the hidden units with indices $b+1, \ldots, k$ to zero. The loss written in the explicit expectation form is,

$$\mathcal{L}_{\mathrm{ND}}(W_1, W_2; X) = \mathop{\mathbb{E}}_{b \sim p_B(\cdot)} \Big[\frac{1}{2n}||X - W_2 \pi_b(W_1 X)||_F^2\Big] \tag{18}$$

In order to derive the analytical form of the expectation, we replace $\pi_b$ in (6) with element-wise masks in the latent space. Let $m_j^{(i)}$ be 0 if the $j^{th}$ latent dimension of the $i^{th}$ data point is dropped out, and 1 otherwise. Define the mask $M \in \{0,1\}^{k \times n}$ as,

$$M = \begin{bmatrix} m_1^{(1)} & \cdots & m_1^{(n)} \\ \vdots & \ddots & \vdots \\ m_k^{(1)} & \cdots & m_k^{(n)} \end{bmatrix}$$

We rewrite (18) as the expectation over $M$ ("$\circ$" denotes element-wise multiplication),

$$\mathcal{L}_{\mathrm{ND}}(W_1, W_2; X) = \mathbb{E}_M \Big[\frac{1}{2n}||X - W_2(M \circ W_1 X)||_F^2\Big] \tag{19}$$

Define $\tilde{X} := W_2(M \circ W_1 X)$. We apply to (19) the bias-variance breakdown of the prediction $\tilde{X}$,

$$\begin{aligned} \mathcal{L}_{\mathrm{ND}}(W_1, W_2; X) &:= \mathbb{E}_M[\mathcal{L}_{\mathrm{ND}}(W_1, W_2, M)] \\ &= \frac{1}{2n}\mathbb{E}[\mathrm{Tr}((X - \tilde{X})(X - \tilde{X})^\top] \\ &= \frac{1}{2n}\mathrm{Tr}(X^\top X - 2X^\top \mathbb{E}[\tilde{X}] + \mathbb{E}[\tilde{X}]^\top \mathbb{E}[\tilde{X}]) \\ &= \frac{1}{2n}\mathrm{Tr}((X - \mathbb{E}[\tilde{X}])^\top(X - \mathbb{E}[\tilde{X}])) + \frac{1}{2}\mathrm{Tr}(\mathrm{Cov}(\tilde{X})) \end{aligned}$$

Define the marginal probability of the latent unit with index $i$ to be kept (not dropped out) as $p_i$,

$$p_i = 1 - \sum_{j=1}^{i-1} p_B(b = j)$$

We also define the matrices $P_D$ and $P_L$ that will be used in the following derivation,

$$P_D = \begin{bmatrix} p_1 & & \\ & \ddots & \\ & & p_k \end{bmatrix}, \quad P_L = \begin{bmatrix} p_1 & p_2 & \cdots & p_k \\ p_2 & p_2 & & p_k \\ \vdots & & & \vdots \\ p_k & p_k & \cdots & p_k \end{bmatrix} \tag{20}$$

We can compute $\mathbb{E}[\tilde{X}]$ and $\mathrm{Tr}(\mathrm{Cov}(\tilde{x}))$ analytically as follows,

$$\mathbb{E}[\tilde{X}] = \mathbb{E}_M[W_2(M \circ W_1 X)] = W_2 P_D W_1 X$$

$$\begin{aligned} \mathrm{Tr}(\mathrm{Cov}(\tilde{x})) &= \frac{1}{n}\mathrm{Tr}(\mathbb{E}[\tilde{X}\tilde{X}^\top]) - \frac{1}{n}\mathrm{Tr}(\mathbb{E}[\tilde{X}]\mathbb{E}[\tilde{X}]^\top) \\ &= \frac{1}{n}\mathrm{Tr}(X^\top W_1^\top (W_2^\top W_2 \circ P_L)W_1 X) - \frac{1}{n}\mathrm{Tr}(X^\top W_1^\top P_D W_2^\top W_2 P_D W_1 X) \end{aligned}$$

Finally, we obtain the analytical form of the expected loss,

$$\begin{aligned} \mathcal{L}_{\mathrm{ND}}(W_1, W_2; X) = &\frac{1}{2n}\mathrm{Tr}(X^\top X) - \frac{1}{n}\mathrm{Tr}(X^\top W_2 P_D W_1 X) \\ &+ \frac{1}{2n}\mathrm{Tr}(X^\top W_1^\top (W_2^\top W_2 \circ P_L)W_1 X) \end{aligned}$$

## D  Conditioning analysis for the deterministic nested dropout

In this section we present an analogous study of the curvature under the Deterministic Nested Dropout objective. We recall from Appendix C that the loss can be written as ($P_D$, $P_L$ as defined in (20)),

$$\mathcal{L}_{\mathrm{ND}}(W_1, W_2; X) = \frac{1}{2n}\mathrm{Tr}(X^\top X) - \frac{1}{n}\mathrm{Tr}(X^\top W_2 P_D W_1 X)$$
$$+ \frac{1}{2n}\mathrm{Tr}(X^\top W_1^\top (W_2^\top W_2 \circ P_L) W_1 X)$$

Let $Q = \mathrm{diag}(q1, \ldots, q_k)$, where $q_i \in \mathbb{R}$, $q_i \neq 0$, for $i = 1, \ldots, k$. The global minima of the objective are not unique, and can be expressed as,

$$W_1^* = QU^\top \tag{21}$$
$$W_2^* = UQ^{-1} \tag{22}$$

We can adopt the same approach as in Appendix B. We will compute quadratic forms with the Hessian of the objective, via paths through the parameter space. We will consider paths along scaling and rotation of the parameters.

**Scaling curvature**  Let $g(\alpha) = \mathcal{L}_{\mathrm{ND}}(W_1^* + \alpha Z_1, W_2^* + \alpha Z_2; X)$. As in Appendix B, we need only compute the second order ($\alpha$) terms in $g(\alpha)$,

$$\alpha^2 [ -\frac{1}{n}\mathrm{Tr}(X^\top Z_2 P_D Z_1 X) + \frac{1}{2n}\mathrm{Tr}(2X^\top Z_1^\top (((W_2^*)^\top Z_2 + Z_2^\top W_2^*) \circ P_L) W_1^* X)$$
$$+ \frac{1}{2n}\mathrm{Tr}(X^\top (W_1^*)^\top (Z_2^\top Z_2 \circ P_L) W_1^* X) + \frac{1}{2n}\mathrm{Tr}(X^\top Z_1^\top ((W_2^*)^\top W_2^* \circ P_L) Z_1 X)] \tag{23}$$

Let $Z = [u_1 \quad 0] \in \mathbb{R}^{m \times k}$, where $u_1 \in \mathbb{R}^m$ is the first column of $U$. Let $Z_1^\top = Z_2 = Z$, we have the following identity,

$$Z^\top Z = U^\top Z = \mathrm{diag}(1, 0, \ldots, 0) \in \mathbb{R}^{k \times k} \tag{24}$$

Substituting (21), (22) into (23), and applying identity (24), the second order term in $g(\alpha)$ becomes,

$$\frac{1}{2}\alpha^2 g''(0) = \alpha^2 \cdot p_1 \sigma_1^2 (1 + \frac{1}{2}(q_1^2 + \frac{1}{q_1^2})) \geq \alpha^2 \cdot 2p_1 \sigma_1^2$$
$$\implies g''(0) \geq 4p_1 \sigma_1^2$$

Applying Lemma 1 and notice that $||Z||_F = 1$, we can get a lower bound for the largest singular value of the Hessian $H$,

$$s_{\max}(H) \geq \frac{\mathrm{vec}([Z_1^\top \quad Z_2])^\top H \mathrm{vec}([Z_1^\top \quad Z_2])}{||[Z_1^\top \quad Z_2]||_F^2} = \frac{g''(0)}{2||Z||_F^2} \geq 2p_1 \sigma_1^2$$

**Rotation curvature**  We use a similar approach as in Appendix B to get a upper bound for the smallest singular value of the Hessian matrix. We consider paths along the (scaled) rotation manifold,

$$W_1 = QR(\theta)Q^{-1}W_1^*$$
$$W_2 = W_2^* QR(\theta)^\top Q^{-1}$$

where $R(\theta)$ is a rotation matrix parameterized by $\theta$, representing the rotation of the $i^{th}$ and $j^{th}$ dimensions only (a Givens rotation).

$$\mathcal{L}_{\mathrm{ND}}(W_1, W_2; X) = \mathrm{Const} - \frac{1}{n}\mathrm{Tr}\Big(X^\top W_2^* QR(\theta)^\top Q^{-1} P_D QR(\theta)Q^{-1}W_1^* X\Big)$$
$$+ \frac{1}{2n}\mathrm{Tr}\Big(X^\top (W_1^*)^\top Q^{-1} R(\theta)^\top Q\Big(Q^{-1}R(\theta)Q(W_2^*)^\top W_2^* QR(\theta)^\top Q^{-1} \circ P_L\Big)QR(\theta)Q^{-1}W_1^* X\Big) \tag{25}$$

Without loss of generality, we consider the loss in the $2 \times 2$ case ($i^{th}$ and $j^{th}$ dimensions only), and denote all terms independent of $\theta$ as $\mathrm{Const}$. Substituting (21) and (22) into (25),

$$\mathcal{L}_{\mathrm{ND}}(W_1, W_2; X) = \mathrm{Const} - \frac{1}{2}\mathrm{Tr}(\begin{bmatrix} \sigma_i^2 & \\ & \sigma_j^2 \end{bmatrix} R(\theta)^\top \begin{bmatrix} p_i & \\ & p_j \end{bmatrix} R(\theta))$$

$$= \mathrm{Const} - \frac{1}{2}[(\sigma_i^2 p_i + \sigma_j^2 p_j)\cos^2\theta + (\sigma_j^2 p_i + \sigma_i^2 p_j)\sin^2\theta]$$

We can compute the derivatives of the objective with respect to $\theta$,

$$\frac{\partial}{\partial\theta}\mathcal{L}_{\mathrm{ND}}(W_1, W_2; X) = \frac{1}{2}(\sigma_i^2 - \sigma_j^2)(p_i - p_j)\sin 2\theta$$

$$\frac{\partial^2}{\partial\theta^2}\mathcal{L}_{\mathrm{ND}}(W_1, W_2; X)\Big|_{\theta=0} = (\sigma_i^2 - \sigma_j^2)(p_i - p_j)\cos 2\theta\Big|_{\theta=0} = (\sigma_i^2 - \sigma_j^2)(p_i - p_j)$$

Also, we compute the Frobenius norm of the path derivative. We use $U_{i,j} \in \mathbb{R}^{m \times 2}$ to denote the matrix containing only the $i^{th}$ and $j^{th}$ columns of $U$.

$$\left\|\frac{d}{d\theta}W_1^\top\right\|_F^2 = \left\|U_{i,j}R(\theta + \frac{\pi}{2})^\top Q\right\|_F^2 = q_i^2 + q_j^2$$

$$\left\|\frac{d}{d\theta}W_2\right\|_F^2 = \left\|U_{i,j}R(\theta + \frac{\pi}{2})^\top Q^{-1}\right\|_F^2 = \frac{1}{q_i^2} + \frac{1}{q_j^2}$$

$$\implies \left\|\frac{d}{d\theta}\begin{bmatrix} W_1^\top & W_2 \end{bmatrix}\right\|_F^2 = q_i^2 + \frac{1}{q_i^2} + q_j^2 + \frac{1}{q_j^2} \geq 4$$

Applying Lemma 1, we obtain an upper bound for the smallest singular value of the Hessian,

$$s_{\min} \leq \frac{\frac{\partial^2}{\partial\theta^2}\mathcal{L}_{\mathrm{ND}}(W_1, W_2; X)\Big|_{\theta=0}}{\left\|\frac{d}{d\theta}\begin{bmatrix} W_1^\top & W_2 \end{bmatrix}\right\|_F^2\Big|_{\theta=0}} \leq \frac{(\sigma_i^2 - \sigma_j^2)(p_i - p_j)}{4}$$

**Conditioning of the objective**   Combining the lower bound of the largest singular value with the upper bound of the smallest singular value of the Hessian matrix, we obtain a lower bound on the condition number,

$$\frac{8p_1\sigma_1^2}{(\sigma_i^2 - \sigma_j^2)(p_i - p_j)}$$

The condition number is controlled by the choice of the cumulative keep probabilities $p_1, \ldots, p_k$. Thus, the condition number can be further lower bounded by the solution of the following optimization problem,

$$\mathrm{cond}(H) \geq \min_{p_1, \ldots, p_k} \max_{i>j} \frac{8p_1\sigma_1^2}{(\sigma_i^2 - \sigma_j^2)(p_i - p_j)}$$

The inner optimization problem can be reduced to a search over a single index $i$, with $j = i + 1$. The minimum of the outer optimization problem is achieved when the inner objective is constant for all $i = 1, \ldots, k - 1$ (otherwise we can adjust $p_1, \ldots, p_k$ to make the inner objective smaller). We denote the constant as $c$, and lower bound it as follows,

$$\frac{1}{c(\sigma_i^2 - \sigma_j^2)} = \frac{(p_i - p_j)}{8p_1\sigma_1^2}, \quad \forall i = 1, \ldots, k - 1$$

$$\implies \frac{1}{c}\sum_{i=1}^{k-1}\frac{1}{\sigma_i^2 - \sigma_j^2} = \sum_{i=1}^{k-1}\frac{(p_i - p_j)}{8p_1\sigma_1^2} = \frac{p_1 - p_k}{8p_1\sigma_1^2} < \frac{1}{8\sigma_1^2}$$

$$\implies c > 8\sigma_1^2\sum_{i=1}^{k-1}\frac{1}{\sigma_i^2 - \sigma_j^2} \geq \frac{8\sigma_1^2(k-1)^2}{\sigma_1^2 - \sigma_k^2}$$

The last inequality is achieved when all terms in the summation are equal. The lower bound of the condition number of the Hessian matrix is,

$$\text{cond}(H) > \frac{8\sigma_1^2(k-1)^2}{\sigma_1^2 - \sigma_k^2}$$

Note that this lower bound will be looser if we do not have the prior knowledge of $\sigma_1, \ldots, \sigma_k$, in order to set $p_1, \ldots, p_k$ appropriately.

## E   Deferred proofs

### E.1   Proof of the Transpose Theorem

The proof of the transpose theorem relied on Lemma 2 (stated below). This result was essentially proved in Kunin et al. [17]. We reproduce the statement and proof here for completeness, which deviates trivially from the original proof.

**Lemma 2.** *The matrix $C = (I - W_2 W_1) X X^\top$ is positive semi-definite at stationary points.*

*Proof.* At stationary points we have,

$$\nabla_{W_2} \mathcal{L}_{\sigma'} = 2(W_2 W_1 - I) X X^T W_1^T + 2 W_2 \Lambda = 0$$

Multiplying on the right by $W_2^\top$ and rearranging gives,

$$X X^\top (W_2 W_1)^\top = W_2 W_1 X X^\top (W_2 W_1)^\top + W_2 \Lambda W_2^\top$$

Both terms on the right are positive definite, thus,

$$X X^\top (W_2 W_1)^\top \succeq W_2 W_1 X X^\top (W_2 W_1)^\top.$$

By Lemma B.1 in Kunin et al. [17], we can cancel $(W_2 W_1)^\top$ on the right[4] and recover $C \succeq 0$.   □

Using Lemma 2, we proceed to prove Theorem 1 (the Transpose Theorem).

*Proof of Theorem 1.* All stationary points must satisfy,

$$\nabla_{W_1} \mathcal{L}_{\sigma'} = \frac{2}{n} W_2^\top (W_2 W_1 - I) X X^\top + 2 \Lambda W_1 = 0$$

$$\nabla_{W_2} \mathcal{L}_{\sigma'} = \frac{2}{n} (W_2 W_1 - I) X X^\top W_1^\top + 2 W_2 \Lambda = 0$$

We have,

$$0 = \nabla_{W_1} \mathcal{L}_{\sigma'} - \nabla_{W_2} \mathcal{L}_{\sigma'}^\top$$
$$= \frac{2}{n}(W_1 - W_2^\top)(I - W_1 W_2) X X^\top + 2\Lambda(W_1 - W_2^\top)$$

By Lemma 2, we know that $C = \frac{1}{n}(I - W_1 W_2) X X^\top$ is positive semi-definite. Further, writing $A = W_1 - W_2^\top$,

$$0 = v^\top A C A^\top v + v^\top \Lambda A A^\top v, \ \forall v$$

As $A C A^\top \succeq 0$, we must have $\forall v$, $v^\top \Lambda A A^\top v \leq 0$. Consider setting $v = e_i$, where $e_i$ is the $i^{th}$ coordinate vector in $\mathbb{R}^k$ ($i^{th}$ entry is 1, and all other entries are 0). We have,

$$e_i^\top \Lambda A A^\top e_i = \lambda_i \|A_i\|_2^2 \leq 0,$$

where $A_i$ denotes the $i^{th}$ row of $A$. Since $\lambda_i > 0$, we have $A_i = 0$. Since this holds for all $i = 1, \ldots, k$, we have $A = 0$.   □

## E.2 Proof of the Landscape Theorem

Before proceeding with our proof of the Landscape Theorem (Theorem 2), we will require the following Lemmas. We begin by proving a weaker version of the landscape theorem (Lemma 3), which allows for symmetry via orthogonal transformations.

$\mathcal{I} \subset \{1, \cdots, m\}$ contains the indices of the learned dimensions. We define $S_{\mathcal{I}}$, $\Lambda_{\mathcal{I}}$, $U_{\mathcal{I}}$ and $I_{\mathcal{I}}$ similarly as in Kunin et al. [17].

- $l = |\mathcal{I}|$. $i_1 < \cdots < i_l$ are increasing indices in $\mathcal{I}$. We use subscript $l$ to denote matrices of dimension $l \times l$.
- $S_{\mathcal{I}} = \mathrm{diag}(\sigma_{i_1}, \ldots, \sigma_{i_l}) \in \mathbb{R}^{l \times l}$, $\Lambda_{\mathcal{I}} = \mathrm{diag}(\lambda_{i_1}, \ldots, \lambda_{i_l}) \in \mathbb{R}^{l \times l}$
- $U_{\mathcal{I}} \in \mathbb{R}^{m \times l}$ has the columns in $U$ with indices $i_1, \ldots, i_l$.
- $I_{\mathcal{I}} \in \mathbb{R}^{m \times l}$ has the columns in the $m \times m$ identity matrix with indices $i_1, \ldots, i_l$.

**Lemma 3** (Weak Landscape Theorem). *All stationary points of* (3) *have the form:*

$$W_1 = O(I_l - \Lambda S_{\mathcal{I}}^{-2})^{\frac{1}{2}} U_{\mathcal{I}}^{\top}$$
$$W_2 = U_{\mathcal{I}}(I_l - \Lambda S_{\mathcal{I}}^{-2})^{\frac{1}{2}} O^{\top}$$

*where $O \in \mathbb{R}^{k \times k}$ is an orthogonal matrix.*

To prove Lemma 3, we introduce Lemma 4 and Lemma 5.

**Lemma 4.** *Given a symmetric matrix $Q \in \mathbb{R}^{m \times m}$, and diagonal matrix $D \in \mathbb{R}^{m \times m}$. If $D$ has distinct diagonal entries, and $Q, D$ satisfy*

$$2QD^2Q = Q^2D^2 + D^2Q^2 \tag{26}$$

*Then $Q$ is diagonal.*

***Proof of Lemma 4.*** We prove Lemma 4 using induction. We use subscript $l$ to denote matrices of dimension $l \times l$.

When $l = 1$, $Q_l$ is trivially diagonal, and Equation (26) always holds.

Assume for some $l \geq 1$, $Q_l$ is diagonal and satisfies (26) for subscript $l$.

We have for dimension $l \times l$:

$$2Q_l D_l^2 Q_l = Q_l^2 D_l^2 + D_l^2 Q_l^2 \tag{27}$$

We write $Q_{l+1}$ and $D_{l+1}^2$ in the following form ($a \in \mathbb{R}^{l \times 1}$, $q, s$ are scalars)

$$Q_{l+1} = \begin{bmatrix} Q_l & a \\ a^{\top} & q \end{bmatrix} \qquad D_{l+1}^2 = \begin{bmatrix} D_l^2 & 0 \\ 0^{\top} & d^2 \end{bmatrix}$$

Expand the LHS and RHS of Equation (26) for subscript $l + 1$:

$$2Q_{l+1}D_{l+1}^2 Q_{l+1} = 2\begin{bmatrix} Q_l & a \\ a^{\top} & q \end{bmatrix}\begin{bmatrix} D_l^2 & 0 \\ 0^{\top} & d^2 \end{bmatrix}\begin{bmatrix} Q_l & a \\ a^{\top} & q \end{bmatrix}$$
$$= 2\begin{bmatrix} Q_l D_l^2 Q_l + d^2 aa^{\top} & Q_l D_l^2 a + d^2 qa \\ a^{\top} D_l^2 Q_l + d^2 qa^{\top} & a^{\top} D_l^2 a + d^2 q^2 \end{bmatrix} \tag{28}$$

$$Q_{l+1}^2 D_{l+1}^2 + D_{l+1}^2 Q_{l+1}^2 = \begin{bmatrix} Q_l & a \\ a^{\top} & q \end{bmatrix}^2 \begin{bmatrix} D_l^2 & 0 \\ 0^{\top} & d^2 \end{bmatrix} + \begin{bmatrix} D_l^2 & 0 \\ 0^{\top} & d^2 \end{bmatrix}\begin{bmatrix} Q_l & a \\ a^{\top} & q \end{bmatrix}^2$$
$$= \begin{bmatrix} \mathrm{RHS}_{1:l,1:l} & \mathrm{RHS}_{1:l,l+1} \\ \mathrm{RHS}_{l+1,1:l} & \mathrm{RHS}_{l+1,l+1} \end{bmatrix}$$

$$\mathrm{RHS}_{1:l,1:l} = Q_l^2 D_l^2 + D_l^2 Q_l^2 + aa^{\top} D_l^2 + D_l^2 aa^{\top} \tag{29}$$

Equate the 1 to $l^{th}$ row and column of LHS and RHS (top-left of Equation (28) and (29)), and apply the induction assumption (27):

$$2d^2 aa^\top = aa^\top D_l^2 + D_l^2 aa^\top$$
$$\implies 0 = aa^\top (D_l^2 - d^2 I) + (D_l^2 - d^2 I)aa^\top$$
$$\implies 0 = 2a_i^2 (s_i^2 - d^2), \ \forall i = 1, \cdots, l, \ D_l^2 = \text{diag}(s_1^2, \cdots, s_l^2)$$

Since $D_{l+1}$ is a diagonal matrix with distinct diagonal entries, $s_i^2 - d^2 \neq 0$ for $\forall i = 1, \cdots, l$. Hence $a = 0$, and $\boldsymbol{Q}_{l+1}$ is diagonal.

It's easy to check that $a = 0$ satisfies Equation (26), hence diagonal $Q_{l+1}$ is a valid solution.

By induction, $Q \in \mathbb{R}^{m \times m}$ is diagonal. $\qquad \square$

**Lemma 5.** *Consider the loss function,*

$$\tilde{L}(Q_1, Q_2) = \text{tr}(Q_2 Q_1 S^2 Q_1^\top Q_2^\top - 2Q_2 Q_1 S^2$$
$$+ 2Q_1 Q_2 \Lambda + S^2)$$

*where $S^2 = \text{diag}(\sigma_1^2, \ldots, \sigma_k^2)$, $\Lambda = \text{diag}(\lambda_1, \ldots, \lambda_k)$ are diagonal matrices with distinct positive elements, and $\forall i = 1, \ldots, k, \sigma_i^2 > \lambda_i$. Then all stationary points satisfying $Q_1^\top = Q_2$ are of the form,*

$$Q_1 = O(I_l - \Lambda_{\mathcal{I}} S_{\mathcal{I}}^{-2})^{\frac{1}{2}} I_{\mathcal{I}}^\top$$

***Proof of Lemma 5.*** Taking derivatives,

$$\frac{\partial \tilde{L}}{\partial Q_1} = 2Q_2^\top Q_2 Q_1 S^2 - 2Q_2^\top S^2 + 2\Lambda^2 Q_2^\top = 0$$

$$\frac{\partial \tilde{L}}{\partial Q_2} = 2Q_2 Q_1 S^2 Q_1^\top - 2S^2 Q_1^\top + 2Q_1^\top \Lambda^2 = 0$$

Multiplying the first equation on the left by $Q_1^\top$, and using $Q_2 = Q_1^\top$, we get,

$$Q_1^\top Q_1 Q_1^\top Q_1 S^2 - Q_1^\top Q_1 S^2 + Q_1^\top \Lambda^2 Q_1 = 0 \qquad (30)$$

Similarly, multiplying the second equation on the right by $Q_1$,

$$Q_1^\top Q_1 S^2 Q_1^\top Q_1 - S^2 Q_1^\top Q_1 + Q_1^\top \Lambda^2 Q_1 = 0$$

Writing $Q = Q_1^\top Q_1$, and equating through $Q_1^\top \Lambda^2 Q_1$,

$$QS^2 Q = Q^2 S^2 + S^2 Q - QS^2$$

Taking the transpose and adding the result,

$$2QS^2 Q = Q^2 S^2 + S^2 Q^2$$

Applying Lemma 4, we have that $Q$ is a diagonal matrix. Following this, $Q$ commutes with both $S^2$ and $\Lambda^2$, thus we can reduce (30) to,

$$Q^2 S^2 = Q(S^2 - \Lambda^2)$$
$$\Rightarrow S^2(S^2 - \Lambda^2)^{-1} Q Q (S^2 - \Lambda^2)^{-1} S^2 = S^2(S^2 - \Lambda^2)^{-1} Q$$

Thus, $S^2(S^2 - \Lambda^2)^{-1} Q$ is idempotent. From here, we can follow the proof of Proposition 4.3 in [17], with the additional use of the transpose theorem, to determine that,

$$Q_1 = O(I_l - \Lambda_{\mathcal{I}}^2 S_{\mathcal{I}}^{-2})^{\frac{1}{2}} I_{\mathcal{I}}^\top$$

$\qquad \square$

**Proof of Weak Landscape Theorem**    We can now proceed with our desired result, the weak landscape theorem.

*Proof of Lemma 3.* Let $Q_1 = W_1 U$, and $Q_2 = U^\top W_2$. We can write the loss as,

$$\mathcal{L}_{\sigma'} = \mathrm{Tr}(Q_2 Q_1 S^2 Q_1^\top Q_2^\top - 2 Q_2 Q_1 S + 2 Q_1 Q_2 \Lambda + S^2) + ||\Lambda^{1/2}(Q_1 - Q_2^\top)||_F^2 \qquad (31)$$

To see this, observe that,

$$||\Lambda^{1/2}(Q_1 - Q_2^\top)||_F^2 = \mathrm{Tr}(Q_1 Q_1^\top \Lambda + Q_2^\top Q_2 \Lambda - 2 Q_1 Q_2 \Lambda)$$
$$= ||\Lambda^{1/2} Q_1||_F^2 + ||Q_2 \Lambda^{1/2}||_F^2 - 2\mathrm{Tr}(Q_1 Q_2 \Lambda)$$

The Transpose Theorem guarantees that the second term in (31) is zero at stationary points. Applying Lemma 5, all stationary points must be of the form:

$$W_1^* = O(I_l - \Lambda_{\mathcal{I}} S_{\mathcal{I}}^{-2})^{\frac{1}{2}} U_{\mathcal{I}}^\top \qquad (32)$$
$$W_2^* = U_{\mathcal{I}}(I_l - \Lambda_{\mathcal{I}} S_{\mathcal{I}}^{-2})^{\frac{1}{2}} O^\top \qquad (33)$$

$\square$

**Proof of the (Strong) Landscape Theorem**    We now present our proof of the strong version of the Landscape Theorem, which removes the orthogonal symmetry present in the weaker version.

*Proof of Theorem 2.* By Theorem 1, at stationary points, $W_1 = W_2^\top$. We write $W_1 = \begin{bmatrix} w_1^\top & w_2^\top & \cdots & w_k^\top \end{bmatrix}^\top$, and $W_2 = \begin{bmatrix} w_1 & w_2 & \cdots & w_k \end{bmatrix}$, where $w_i$ for $i = 1, \cdots, k$ is the $i^{th}$ column of the decoder.

Define the regularization term in the loss as $\psi(W_1, W_2)$.

$$\psi(W_1, W_2) = ||\Lambda^{1/2} W_1||_F^2 + ||W_2 \Lambda^{1/2}||_F^2 = 2||\Lambda^{1/2} W_1||_F^2$$

Let $\tilde{W}_1 = R_{ij} W_1$ and $\tilde{W}_2 = W_2 R_{ij}^\top$, where $R_{ij}$ is the rotational matrix for the $i^{th}$ and $j^{th}$ components.

$$R_{ij} = \begin{bmatrix} 1 & & & & & & \\ & \ddots & & & & & \\ & & \cos\theta & & -\sin\theta & & \\ & & & \ddots & & & \\ & & \sin\theta & & \cos\theta & & \\ & & & & & \ddots & \\ & & & & & & 1 \end{bmatrix}$$

$$\psi(\tilde{W}_1, \tilde{W}_2) = ||\Lambda^{1/2}\tilde{W}_1||_2^2 + ||\tilde{W}_2\Lambda^{1/2}||_2^2$$
$$= \text{Tr}(\Lambda^{1/2}\tilde{W}_1\tilde{W}_1^\top\Lambda^{1/2}) + \text{Tr}(\Lambda^{1/2}\tilde{W}_2^\top\tilde{W}_2\Lambda^{1/2})$$
$$= \text{Tr}(\Lambda^{1/2}R_{ij}W_1W_1^\top R_{ij}^\top\Lambda^{1/2}) + \text{Tr}(\Lambda^{1/2}R_{ij}W_2^\top W_2 R_{ij}^\top\Lambda^{1/2})$$

$$= 2\text{Tr}(\Lambda^{1/2}
\begin{bmatrix}
w_1^\top \\
\vdots \\
w_i^\top\cos\theta - w_j^\top\sin\theta \\
\vdots \\
w_i^\top\sin\theta + w_j^\top\cos\theta \\
\vdots \\
w_k^\top
\end{bmatrix}
\begin{bmatrix}
w_1^\top \\
\vdots \\
w_i^\top\cos\theta - w_j^\top\sin\theta \\
\vdots \\
w_i^\top\sin\theta + w_j^\top\cos\theta \\
\vdots \\
w_k^\top
\end{bmatrix}^\top
\Lambda^{1/2})$$

$$= 2\text{Tr}(
\begin{bmatrix}
\lambda_1^{1/2}w_1^\top \\
\vdots \\
\lambda_i^{1/2}(w_i^\top\cos\theta - w_j^\top\sin\theta) \\
\vdots \\
\lambda_j^{1/2}(w_i^\top\sin\theta + w_j^\top\cos\theta) \\
\vdots \\
\lambda_k^{1/2}w_k^\top
\end{bmatrix}
\begin{bmatrix}
\lambda_1^{1/2}w_1^\top \\
\vdots \\
\lambda_i^{1/2}(w_i^\top\cos\theta - w_j^\top\sin\theta) \\
\vdots \\
\lambda_j^{1/2}(w_i^\top\sin\theta + w_j^\top\cos\theta) \\
\vdots \\
\lambda_k^{1/2}w_k^\top
\end{bmatrix}^\top
)$$

$$= 2[\lambda_i(w_i^\top\cos\theta - w_j^\top\sin\theta)^\top(w_i^\top\cos\theta - w_j^\top\sin\theta)$$
$$+ \lambda_j(w_i^\top\sin\theta + w_j^\top\cos\theta)^\top(w_i^\top\sin\theta + w_j^\top\cos\theta)) + \sum_{l=1,l\neq i,i\neq j}^{k}\lambda_l w_l^\top w_l]$$
$$= 2[(\lambda_i w_i^\top w_i + \lambda_j w_j^\top w_j)\cos^2\theta + (\lambda_j w_i^\top w_i + \lambda_i w_j^\top w_j)\sin^2\theta$$
$$+ 4(\lambda_j - \lambda_i)w_i^\top w_j\sin\theta\cos\theta + \sum_{l=1,l\neq i,i\neq j}^{k}\lambda_l w_l^\top w_l]$$
$$= 2[A\cos(2\theta + B) + C + \sum_{l=1,l\neq i,i\neq j}^{k}\lambda_l w_l^\top w_l]$$

Where $A, B, C$ satisfy:

$$A\cos B = \frac{1}{2}(\lambda_j - \lambda_i)(w_j^\top w_j - w_i^\top w_i) \tag{34}$$
$$A\sin B = -2(\lambda_j - \lambda_i)w_i^\top w_j \tag{35}$$

In order for $\psi(\tilde{W}_1, \tilde{W}_2)$ to be a stationary point at $\theta = 0$, we need either of the two necessary conditions to be true for $\forall i < j$:

Condition 1: $A = 0 \iff w_i^\top w_j = 0$ and $w_i^\top w_i = w_j^\top w_j$

Condition 2: $A \neq 0$ and $B = \beta\pi, \ \beta \in \mathbb{Z} \iff w_i^\top w_j = 0$ and $w_j^\top w_j \neq w_i^\top w_i$

The two conditions can be consolidated to one, i.e. the columns of the decoder needs to be orthogonal.

$$\forall i, j \in \{1, \cdots, k\}, \quad w_i^\top w_j = 0$$

The following Lemma uses such orthogonality to constrain the form that the matrix $O$ in (32) and (33) can take.

**Lemma 6.** *Let $W_1^*$, $W_2^*$ be in the form of (32) and (33). And let $W_1^* = \begin{bmatrix} w_1^\top & w_2^\top & \cdots & w_k^\top \end{bmatrix}^\top$, and $W_2^* = \begin{bmatrix} w_1 & w_2 & \cdots & w_k \end{bmatrix}$, where $w_i \in \mathbb{R}^m$ for $i = 1, \cdots, k$ is the $i^{th}$ columns of the $W_2^*$.*

*If for $\forall i, j \in \{1, \cdots, k\}$, $w_i^\top w_j = 0$, then $O$ has exactly one entry of $\pm 1$ in each row and at most one entry of $\pm 1$ in each column, and zeros elsewhere.*

*Proof of Lemma 6.*

$$(W_2^*)^\top W_2^* = O(I_l - \Lambda S_{\mathcal{I}}^{-2})^{\frac{1}{2}} U_{\mathcal{I}}^\top U_{\mathcal{I}} (I_l^2 - \Lambda S_{\mathcal{I}}^{-2})^{\frac{1}{2}} O^\top = O(I_l - \Lambda S_{\mathcal{I}}^{-2}) O^\top \qquad (36)$$

Note that $(I_l - \Lambda S_{\mathcal{I}}^{-2})$ is a diagonal matrix with strictly descending positive diagonal entries, so (36) is an SVD to $(W_2^*)^\top W_2^*$.

Because $W_2^*$ has orthogonal columns, $(W_2^*)^\top W_2^*$ is a diagonal matrix. There exists a permutation matrix $P_0 \in \mathbb{R}^{k \times k}$, such that $W_2^* P_0^\top$ has columns ordered strictly in descending magnitude. Let $\bar{W}_2^* = W_2^* P_0^\top$, and $\bar{O} = P_0 O$, then

$$
\begin{aligned}
(\bar{W}_2^*)^\top \bar{W}_2^* &= (W_2^* P_0^\top)^\top W_2^* P_0^\top \\
&= P_0 O(I_l - \Lambda S_{\mathcal{I}}^{-2}) O^\top P_0^\top \\
&= \bar{O}(I_l - \Lambda S_{\mathcal{I}}^{-2}) \bar{O}^\top \qquad (37) \\
&= I_l - \Lambda S_{\mathcal{I}}^{-2} \qquad (38)
\end{aligned}
$$

Note that $\bar{O} = P_0 O$ also have orthonormal columns, we have $\bar{O}^\top \bar{O} = I$. Let $\bar{O} = \begin{bmatrix} o_1^\top & o_2^\top & \cdots & o_k^\top \end{bmatrix}^\top$, where $o_j \in \mathbb{R}^{1 \times l}$ are rows of $O$. From (37) and (38), we have for $i \in \{1, \cdots, l\}, j \in \{1, \cdots, k\}$:

$$
\begin{aligned}
&\bar{O}^\top (I_l - \Lambda S_{\mathcal{I}}^{-2}) = (I_l - \Lambda S_{\mathcal{I}}^{-2}) \bar{O}^\top \\
\implies\ &(\bar{O}^\top (I_l - \Lambda S_{\mathcal{I}}^{-2}))_{ij} = ((I_l - \Lambda S_{\mathcal{I}}^{-2}) \bar{O}^\top)_{ij} \quad \forall i, j \in \{1, \cdots, k\} \\
\implies\ &(o_j)_i (1 - \lambda_j \sigma_{i_j}^{-2}) = (o_j)_i (1 - \lambda_i \sigma_{i_i}^{-2}) \\
\implies\ &(o_j)_i (\lambda_i \sigma_{i_i}^{-2} - \lambda_j \sigma_{i_j}^{-2}) = 0
\end{aligned}
$$

Since $(I_l - \Lambda S_{\mathcal{I}}^{-2})$ is a diagonal matrix with strictly descending entries, we have $\lambda_i \sigma_{i_i}^{-2} - \lambda_j \sigma_{i_j}^{-2} \neq 0$ for $i \neq j$. Hence $(o_j)_i = 0$ for $i \neq j$, i.e. $\bar{O}$ is diagonal. Since $\bar{O}$ has orthonormal columns, it has diagonal entries $\pm 1$.

$$O = P_0^{-1} \bar{O} = P_0^\top \bar{O}$$

Therefore, $O$ has exactly one entry of $\pm 1$ in each row, and at most one entry of $\pm 1$ in each column, and zeros elsewhere. $\qquad \square$

We now finish the proof for Theorem 2. Applying Lemma 6, we can rewrite the stationary points using rank $k$ matrices $S$ and $U$:

$$
\begin{aligned}
W_1^* &= P(I - \Lambda S^{-2})^{\frac{1}{2}} U^T \\
W_2^* &= U(I - \Lambda S^{-2})^{\frac{1}{2}} P
\end{aligned}
$$

Where $P \in \mathbb{R}^{k \times k}$ has exactly one $\pm 1$ in each row and each column with index in $\mathcal{I}$, and zeros elsewhere. This concludes the proof.

$\qquad \square$

### E.3 Proof of recovery of ordered, axis-aligned solution at global minima

**Lemma 7** (Global minima – necessary condition 1). *Let the encoder $(W_1^*)$ and decoder $(W_2^*)$ of the non-uniform $\ell_2$ regularized LAE have the form in (4) and (5). If $0 < \lambda_i < \sigma_i^2$ for $\forall i = 1, \cdots, k$, then $(W_1^*, W_2^*)$ can be at global minima only if $P$ has full rank.*

*Proof of Lemma 7.* We prove the contrapositive: if $\mathrm{rank}(P) < k$, then $(W_1^*, W_2^*)$ in (4) and (5) is not at global minimum.

Since $\mathrm{rank}(P) < k$, there exists a matrix $\delta P \in \mathbb{R}^{k \times k}$ such that $\delta P$ has all but one element equal to 0, and $\delta P_{ij} = h > 0$, for some $i, j \in \{1, \dots, k\}$, where the $i^{th}$ row and $j^{th}$ column of $P$ are all zeros.

$$\delta W_1 = \delta P (I - \Lambda S^{-2})^{\frac{1}{2}} U^T$$

$$\delta W_2 = U(I - \Lambda S^{-2})^{\frac{1}{2}} \delta P^\top$$

$$
\begin{aligned}
&\mathcal{L}_{\sigma'}(W_1^* + \delta W_1, W_2^* + \delta W_2) \\
&= \frac{1}{n} \|X - (W_2^* + \delta W_2)(W_1^* + \delta W_1)X\|_F^2 \\
&\quad + \|\Lambda^{1/2}(W_1^* + \delta W_1)\|_F^2 + \|(W_2^* + \delta W_2)\Lambda^{1/2}\|_F^2 \\
&= \frac{1}{n}\mathrm{Tr}((I - (W_2^* + \delta W_2)(W_1^* + \delta W_1))XX^\top(I - (W_2^* + \delta W_2)(W_1^* + \delta W_1))) \\
&\quad + \mathrm{Tr}(\Lambda^{1/2}(W_1^* + \delta W_1)(W_1^* + \delta W_1)^\top \Lambda^{1/2}) + \mathrm{Tr}(\Lambda^{1/2}(W_2^* + \delta W_2)^\top(W_2^* + \delta W_2)\Lambda^{1/2}) \\
&= \mathrm{Tr}((I - (I - \Lambda S^{-2})(P + \delta P)^\top(P + \delta P))^2 S^2) \\
&\quad + 2\mathrm{Tr}(\Lambda(P + \delta P)(I - \Lambda S^{-2})(P + \delta P)^\top) \\
&= \mathcal{L}_{\sigma'}(W_1^*, W_2^*) + [(1 - (1 - \lambda_i \sigma_i^{-2})h^2)^2 - 1]\sigma_i^2 + 2\lambda_i(1 - \lambda_i\sigma_i)^{-2})h^2 \\
&= \mathcal{L}_{\sigma'}(W_1^*, W_2^*) - 2(\sigma_i^2 - \lambda_i)(1 - \lambda_i\sigma_i^{-2})h^2 + (1 - \lambda_i\sigma_i^{-2})^2\sigma_i^2 h^4 \\
&= \mathcal{L}_{\sigma'}(W_1^* - \delta W_1, W_2^* - \delta W_2)
\end{aligned}
$$

The first derivative of $(W_1^*, W_2^*)$ along $(\delta W_1, \delta W_2)$ is zero:

$$
\begin{aligned}
&\lim_{h \to 0} \frac{\mathcal{L}_{\sigma'}(W_1^* + \delta W_1, W_2^* + \delta W_2) - \mathcal{L}_{\sigma'}(W_1^*, W_2^*)}{h} \\
&= \lim_{h \to 0} \frac{-2(\sigma_i^2 - \lambda_i)(1 - \lambda_i\sigma_i^{-2})h^2 + (1 - \lambda_i\sigma_i^{-2})^2\sigma_i^2 h^4}{h} \\
&= 0
\end{aligned}
$$

The second derivative of $(W_1^*, W_2^*)$ along $(\delta W_1, \delta W_2)$ is negative (note that $0 < \lambda_i < \sigma_i^2$):

$$
\begin{aligned}
&\lim_{h \to 0} \frac{\mathcal{L}_{\sigma'}(W_1^* + \delta W_1, W_2^* + \delta W_2) - 2\mathcal{L}_{\sigma'}(W_1^*, W_2^*) + \mathcal{L}_{\sigma'}(W_1^* - \delta W_1, W_2^* - \delta W_2)}{h^2} \\
&= \lim_{h \to 0} \frac{2\mathcal{L}_{\sigma'}(W_1^* + \delta W_1, W_2^* + \delta W_2) - 2\mathcal{L}_{\sigma'}(W_1^*, W_2^*)}{h^2} \\
&= \lim_{h \to 0} 2\frac{-2(\sigma_i^2 - \lambda_i)(1 - \lambda_i\sigma_i^{-2})h^2 + (1 - \lambda_i\sigma_i^{-2})^2\sigma_i^2 h^4}{h^2} \\
&= -4(\sigma_i^2 - \lambda_i)(1 - \lambda_i\sigma_i^{-2}) \\
&< 0
\end{aligned}
$$

Therefore, if $\mathrm{rank}(P) < k$, $(W_1^*, W_2^*)$ is not at global minima. The contrapositive states that if $(W_1^*, W_2^*)$ is at global minima, then $P$ has full rank. $\square$

**Lemma 8** (Global minima – necessary condition 2). *Let the encoder $(W_1^*)$ and decoder $(W_2^*)$ of the non-uniform $\ell_2$ regularized LAE have the form in (4) and (5), and $P$ has full rank. Then $(W_1^*, W_2^*)$ can be at global minimum only if $P$ is diagonal.*

*Proof of Lemma 8.* Following similar analysis for the proof of Theorem 2, we have (34) and (35). In order for $\theta = 0$ to be a global optimum, it must be a local optimum. Therefore, for $\forall i < j$, we need either of the following necessary conditions to be true:

Condition 1: $A = 0 \iff w_i^\top w_j = 0$ and $w_i^\top w_i = w_j^\top w_j$

Condition 2: $A\cos B < 0$ and $B = \beta\pi, \ \beta \in \mathbb{Z} \iff w_i^\top w_j = 0$ and $w_i^\top w_i > w_j^\top w_j$

The two conditions can be consolidated to the following ($i < j$):

$$w_i^\top w_j = 0 \ \text{ and } \ w_i^\top w_i \geq w_j^\top w_j$$

Then, $(W_2^*)^\top (W_2^*)$ is a diagonal matrix with non-negative diagonal entries sorted in descending order.

$$(W_2^*)^\top (W_2^*) = P(I_l - \Lambda S_{\mathcal{I}}^{-2}) P^\top \tag{39}$$

Since the diagonal entries of $(I_l - \Lambda S_{\mathcal{I}}^{-2})$ are positive and sorted in strict descending order, and that (39) is an SVD of $(W_2^*)^\top (W_2^*)$, we have:

$$(W_2^*)^\top (W_2^*) = (I_l - \Lambda S_{\mathcal{I}}^{-2})$$

We can use the same technique as the proof of Lemma 6 to prove that $P$ must be diagonal. $\square$

**Lemma 9** (Global minima – sufficient condition). *Let $\bar{I} \in \mathbb{R}^{k \times k}$ be a diagonal matrix with diagonal elements equal to $\pm 1$. The encoder ($W_1^*$) and decoder ($W_2^*$) of the following form are at global minima of the non-uniform $\ell_2$ LAE objective.*

$$W_1^* = \bar{I}(I - \Lambda S^{-2})^{\frac{1}{2}} U^T \tag{40}$$

$$W_2^* = U(I - \Lambda S^{-2})^{\frac{1}{2}} \bar{I} \tag{41}$$

*Proof of Lemma 9.* Because the objective of the non-uniform regularized LAE is differentiable everywhere for $W_1$ and $W_2$, all local minima (therefore also global minima) must occur at stationary points. Theorem 2 shows that the stationary points must be of the form (4) and (5). Lemma 7 further shows that a necessary condition for the global minima is when $l = k$, i.e. the encoder and decoder must be of the form in (40) and (41).

In order to prove that (40) and (41) are sufficient condition for global minima, it is sufficient to show that all $W_1^*$, $W_2^*$ that satisfy (40) and (41) (i.e. all $\bar{I}$) result in the same loss. Notice that $\bar{I}^2 = I$, then:

$$
\begin{aligned}
\mathcal{L}_{\sigma'}(W_1^*, W_2^*) &= \frac{1}{n} \|X - W_2^* W_1^* X\|_F^2 + \|\Lambda^{1/2} W_1^*\|_F^2 + \|W_2^* \Lambda^{1/2}\|_F^2 \\
&= \frac{1}{n} \|X - W_2^* W_1^* X\|_F^2 + \mathrm{Tr}(\Lambda^{1/2} W_1^* (W_1^*)^\top \Lambda^{1/2}) \\
&\quad + \mathrm{Tr}(\Lambda^{1/2} (W_2^*)^\top W_2^* \Lambda^{1/2}) \\
&= \frac{1}{n} \|X - U(I - \Lambda S^{-2})^{\frac{1}{2}} \bar{I}^2 (I - \Lambda S^{-2})^{\frac{1}{2}} U^T X\|_F^2 \\
&\quad + 2\mathrm{Tr}(\Lambda^{1/2} \bar{I}(I - \Lambda S^{-2})^{\frac{1}{2}} U^T U (I - \Lambda S^{-2})^{\frac{1}{2}} \bar{I}^\top \Lambda^{1/2}) \\
&= \frac{1}{n} \|X - U(I - \Lambda S^{-2}) U^T X\|_F^2 + 2\mathrm{Tr}(\Lambda(I - \Lambda S^{-2}))
\end{aligned}
\tag{42}
$$

According to (42), $\mathcal{L}_{\sigma'}(W_1^*, W_2^*)$ is constant with respect to $\bar{I}$. Hence, (40) and (41) are sufficient conditions for global minima of the non-uniform $\ell_2$ regularized LAE objective. $\square$

***Proof of Theorem 3.*** From Lemma 7, 8, and 9, we conclude that the global minima of the non-uniform $\ell_2$ regularized LAE are achieved if and only if the encoder ($W_1^*$) and decoder ($W_2^*$) are of the form in (40) and (41), i.e. ordered, axis-aligned individual principal component directions.

We have proven in Lemma 7 that for $l < k$, there exists a direction for which the second derivative of the objective is negative. We have proven also that stationary points with $l = k$ are either global optima, or saddle points (Lemma 8, 9). Hence, there do not exist local minima that are not global minima. $\square$

### E.4 Proof of local linear convergence of RAG

*Proof of Theorem 5.* Applying Assumption 1, the instantaneous update for RAG is,

$$\dot{W}_1 = \frac{1}{n}AW_1$$

$$\dot{W}_2 = \frac{1}{n}W_2 A$$

The instantaneous update for $YY^\top$ is,

$$\frac{d}{dt}(YY^\top) = \frac{1}{n}(AYY^\top + YY^\top A^\top)$$

Let $y_{ij}$ be the $i, j^{th}$ element of $YY^\top$, and $i < j$, then,

$$\frac{d}{dt}y_{ii} = \frac{2}{n}\left(-\sum_{l=1}^{i-1} y_{il}^2 + \sum_{l=i+1}^{k} y_{il}^2\right)$$

$$\frac{d}{dt}y_{ij} = -\frac{1}{n}(y_{ii} - y_{jj})y_{ij} + \frac{2}{n}\left(-\sum_{l=1}^{i-1} y_{il}y_{jl} + \sum_{l=j+1}^{k} y_{il}y_{jl}\right) \qquad (43)$$

With Assumption 2, we can write (43) as:

$$\frac{d}{dt}y_{ij} = -\frac{1}{n}(y_{ii} - y_{jj})y_{ij} + \mathcal{O}(\frac{\epsilon^2}{k}) \qquad (44)$$

The first term in (44) collects the products of diagonal and off-diagonal elements, and is of order $\mathcal{O}(\frac{\epsilon}{k})$. The second term in (44) collects second-order off-diagonal terms. With $0 < \epsilon \ll 1$, we can drop the second term.

Also, applying Assumption 3, we have $y_{ii} > y_{jj}$.

$$\frac{d}{dt}|y_{ij}| \approx -\frac{1}{n}(y_{ii} - y_{jj})|y_{ij}|$$

The instantaneous change of the "non-diagonality" $Nd(\frac{1}{n}YY^\top)$ is,

$$\frac{d}{dt}Nd(\frac{1}{n}YY^\top) = \frac{d}{dt}\left(2\sum_{i=1}^{k-1}\sum_{j=i+1}^{k}\frac{1}{n}|y_{ij}|\right) = 2\sum_{i=1}^{k-1}\sum_{j=i+1}^{k}\frac{1}{n}\left(\frac{d}{dt}|y_{ij}|\right)$$

$$\approx 2\sum_{i=1}^{k-1}\sum_{j=i+1}^{k}\frac{1}{n}\left(-\frac{1}{n}(y_{ii} - y_{jj})|y_{ij}|\right)$$

$$\leq -g \cdot \left(2\sum_{i=1}^{k-1}\sum_{j=i+1}^{k}\frac{1}{n}|y_{ij}|\right)$$

$$= -g \cdot Nd(\frac{1}{n}YY^\top)$$

Hence, $Nd(\frac{1}{n}YY^\top)$ converges to 0 with an instantaneous linear rate of $g$. $\qquad \square$

### E.5 Convergence of latent space rotation to axis-aligned solutions

We first state LaSalle's invariance principle [13] in Lemma 10, which is used in Theorem 4 to prove the convergence of latent space rotation to the set of axis-aligned solutions.

**Lemma 10** (LaSalle's invariance principle (local version))**.** *Given dynamical system $\dot{x} = f(x)$ where $x$ is a vector of variables, and $f(x^*) = 0$. If a continuous and differentiable real-valued function $V(x)$ satisfies,*

$$\dot{V}(x) \leq 0 \text{ for } \forall x$$

*Then $\dot{V}(x) \to 0$ as $t \to \infty$.*

*Moreover, if there exists a neighbourhood $N$ of $x^*$ such that for $x \in N$,*

$$V(x) > 0 \text{ if } x \neq x^*$$

*And,*

$$\dot{V}(x) = 0 \; \forall \, t \geq 0 \implies x(t) = x^* \; \forall \, t \geq 0$$

*Then $x^*$ is locally asymptotically stable.*

In Section 6.3, we gave an informal statement of Theorem 4. Here, we state the theorem formally.

**Theorem 4** (Global convergence to axis-aligned solutions)**.** Let $O_0 \in \mathbb{R}^{k \times k}$ be an orthogonal matrix, $W \in \mathbb{R}^{k \times m}$ ($k < m$). $X$ and $U$ are as defined in Section 2. $\triangledown(\cdot)$ and $\vartriangle(\cdot)$ are as defined in Algorithm 1. Consider the following dynamical system,

$$\dot{W} = \frac{1}{2n}(\triangledown(WXXW^\top) - \vartriangle(WXXW^\top))W \tag{45}$$

$$W(0) = O_0 U^\top \tag{46}$$

Then $W(t) \to PU^\top$ as $t \to \infty$, where $P \in \mathbb{R}^{k \times k}$ is a permutation matrix with non-zero elements $\pm 1$. Also, the dynamical system is asymptotically stable at $\tilde{I}U^\top$, where $\tilde{I}$ is a diagonal matrix with diagonal entries $\pm 1$.

It is straightforward to show that (45) and (46) are equivalent to the instantaneous limit of RAG on the orthogonal subspace $W_1 = W_2^\top = OU^\top$ ($O$ is an orthogonal matrix). To see this, notice that on the orthogonal subspace, the gradient of $W_1$ and $W_2$ with respect to the reconstruction loss are zero,

$$\nabla_{W_1}\mathcal{L}(W_1 = OU^\top, W_2 = UO^\top; X) = 0$$

$$\nabla_{W_2}\mathcal{L}(W_1 = OU^\top, W_2 = UO^\top; X) = 0$$

Theorem 4 states that in the instantaneous limit, an LAE that is initialized on the orthogonal subspace and is updated by Algorithm 1 globally converges to the set of axis-aligned solutions. Moreover, the convergence to the set of *ordered* axis-aligned solutions is asymptotically stable. We provide the proof below.

*Proof.* We first show that $W(t)$ remains on the orthogonal subspace, i.e. $W(t) = O(t)U^\top$ for $\forall \, t$, where $O(t)$ is orthogonal. To reduce the notation clutter, we define $A(W) = \frac{1}{2n}(\triangledown(WXXW^\top) - \triangledown(WXXW^\top))$. We take the time derivative of $WW^\top$,

$$\frac{d(WW^\top)}{dt} = \dot{W}W^\top + W\dot{W}^\top = A(W)WW^\top + WW^\top A(W)^\top = A(W)WW^\top - WW^\top A(W)$$

The last inequality follows from the observation that $A(W)$ is skew-symmetric, so that $A(W)^\top = -A(W)$. Since $W(0)W(0)^\top = I$, and $WW^\top = I \implies \frac{d(WW^\top)}{dt} = 0$, we have,

$$W(t)W(t)^\top = I \text{ for } \forall \, t \geq 0$$

From the dynamical equation (45), we know that $W(t)$ has the form $W(t) = G(t)U^\top$ for some matrix $G(t) \in \mathbb{R}^{k \times k}$. We have,

$$W(t)W(t)^\top = G(t)U^\top U G(t)^\top = G(t)G(t)^\top = I \implies G(t) \text{ is orthogonal.}$$

We move on to use LaSalle's invariance principle to prove Theorem 4. The rest of the proof is divided into two parts. In the first part, we prove that $W(t) \to PU^\top$ as $t \to \infty$, i.e. $W(t)$ globally converges to axis-aligned solutions. In the second part, we prove that the ordered, axis-aligned solution $\tilde{I}U^\top$ is locally asymptotically stable.

Let $\Sigma = \frac{1}{n}XX^\top$. We define $V(W)$ as,

$$V(W) = \mathrm{Tr}((S^2 - W\Sigma W^\top)D) \tag{47}$$

Where $S$ is as defined in Section 2, and $D = \mathrm{diag}(d_1, \ldots, d_k)$, with $d_1 > \cdots > d_k > 0$.

Note that definition (47) is the Brockett cost function [1] with an offset. The Brockett cost function achieves minimum when the rows of $W$ are the eigenvectors of $\Sigma$. See Appendix G for a detailed discussion of the connection between the rotation augmented gradient and the Brockett cost function.

**Part 1 (global convergence to axis-aligned solutions)** In this part, we compute $\dot{V}(W)$, and invoke the first part of LaSalle's invariance principle to show global convergence to axis-aligned solutions.

Denote the (transposed) $i^{th}$ row of $W$ as $w_i \in \mathbb{R}^{m \times 1}$. We rewrite (45) in terms of rows of $W$,

$$\dot{w}_i = -\frac{1}{2} \sum_{j=1}^{i-1} (w_i^\top \Sigma w_j) w_j + \frac{1}{2} \sum_{j=i+1}^{k} (w_i^\top \Sigma w_j) w_j$$

We proceed to compute $\dot{V}(W)$,

$$\dot{V}(W) = -2 \sum_{i=1}^{k} d_i w_i^\top \Sigma \dot{w}_i = \sum_{i=1}^{k} d_i \left[ \sum_{j=1}^{i-1} (w_i^\top \Sigma w_j)^2 - \sum_{j=i+1}^{k} (w_i^\top \Sigma w_j)^2 \right]$$

$$= \sum_{i=2}^{k} \sum_{j=1}^{i-1} d_i (w_i^\top \Sigma w_j)^2 - \sum_{i=1}^{k-1} \sum_{j=i+1}^{k} d_i (w_i^\top \Sigma w_j)^2$$

$$= \sum_{i=2}^{k} \sum_{j=1}^{i-1} d_i (w_i^\top \Sigma w_j)^2 - \sum_{j=1}^{k-1} \sum_{i=j+1}^{k} d_j (w_i^\top \Sigma w_j)^2$$

$$= \sum_{i=2}^{k} \sum_{j=1}^{i-1} d_i (w_i^\top \Sigma w_j)^2 - \sum_{i=2}^{k} \sum_{j=1}^{i-1} d_j (w_i^\top \Sigma w_j)^2$$

$$= \sum_{i=2}^{k} \sum_{j=1}^{i-1} (d_i - d_j)(w_i^\top \Sigma w_j)^2$$

Since $d_i < d_j$ for $\forall\, i > j$, we have,

$$\dot{V}(W) \le 0 \tag{48}$$

The equality in (48) holds if and only if $\forall\, i \ne j$, $w_i^\top \Sigma w_j = 0$, or, written in matrix form, $WXX^\top W^\top$ is diagonal.

$$\dot{V}(W) = 0 \iff WXX^\top W^\top \text{ is diagonal} \tag{49}$$

Since we also have $W = OU^\top$, and using the SVD of $X$, we can see that (49) is equivalent to,

$$W = PU^\top$$

Also, $W = PU^\top$ are stationary points of the dynamical equation (45). By LaSalle's invariance principle, we have,

$$\dot{V}(W) \to 0 \text{ as } t \to \infty \implies W(t) \to PU^\top \text{ as } t \to \infty$$

$W(t)$ globally converges to the set of axis-aligned solutions. This concludes the first part of the proof.

**Part 2 (asymptotic convergence to optimal representation)** We break down this part of the proof into two steps. First, we show that $V(W)$ is positive definite locally at $\tilde{I}U^\top$. Then, we show that $\tilde{I}U^\top$ is the only solution to $\dot{V}(W) = 0$ in its neighbourhood.

We first show that $V(W)$ is positive definite at $W = \tilde{I}U^\top$. Note that columns of $U$ contain the ordered left singular vectors of $X$. We can rewrite (47) as,

$$V(W) = -\text{Tr}(OS^2O^\top D) + \sum_{i=1}^{k} d_i \sigma_i^2 = -\sum_{i=1}^{k} \sum_{j=1}^{k} d_i \sigma_j^2 O_{ij}^2 + \sum_{i=1}^{k} d_i \sigma_i^2 \tag{50}$$

We use $O_{ij}$ to denote the component with row and column index $i$, $j$ respectively. (50) is minimized when $O = \tilde{I}$ and takes value zero. It is positive everywhere else, and thus, $V(W)$ is positive definite at $W = \tilde{I}U^\top$.

Now, we show that $W = \tilde{I}U^\top$ is the only solution to $\dot{V}(W) = 0$ within some neighbourhood around itself. Since permutation matrices $P$ are finite and distinct, we can find a neighbourhood around each $\tilde{I}$ on the Stiefel manifold $V_k(\mathbb{R}^k)$, in which $W = \tilde{I}U^\top$ is the unique solution for $\dot{V}(W)$. We mathematically state this below,

$$\exists \text{ some neighbourhood } N \text{ on } V_k(\mathbb{R}^k) \text{ around } \tilde{I}, \text{ such that}$$
$$\left[O \in N, \ \dot{V}(OU^\top) = 0 \ \forall \, t \geq 0\right] \implies O = \tilde{I}$$

This means that local to $W = \tilde{I}U^\top$, $\dot{V}(W) = 0$ for $\forall \, t \geq 0$ implies $W = \tilde{I}U^\top$.

We have satisfied all the necessary conditions to invoke LaSalle's invariance principle. Thus, $W = \tilde{I}U^\top$ is locally asymptotically stable. $\qquad\square$

# F    Connection of non-uniform $\ell_2$ regularization to linear VAE with diagonal covariance

Consider the following VAE model,

$$p(x|z) = \mathcal{N}(Wz + \mu, \sigma^2 I)$$
$$q(z|x) = \mathcal{N}(V(x - \mu), D)$$

Where $W$ is the decoder, $V$ is the encoder, and $D$ is the diagonal covariance matrix. The ELBO objective is,

$$\text{ELBO} = -KL(q(z|x)||p(z)) + \mathbb{E}_{q(z|x)}[\log p(x|z)]$$

It's shown in [19] that such a linear VAE with diagonal latent covariance can learn axis-aligned principal component directions. We show in this section that training such a linear VAE with ELBO is closely related to training a non-uniform $\ell_2$ regularized LAE.

As derived in Appendix C.2 of [19], the gradients of the ELBO with respect to $D, V$ and $W$, are,

$$\nabla D = \frac{n}{2}(D^{-1} - I - \frac{1}{\sigma^2}\text{diag}(W^\top W))$$
$$\nabla V = \frac{n}{\sigma^2}(W^\top - (W^\top W + \sigma^2 I)V)\Sigma$$
$$\nabla W = \frac{n}{\sigma^2}(\Sigma V^\top - DW - WV\Sigma V^\top)$$

Where $\Sigma = \frac{1}{n}XX^\top$. The optimal $D^* = \sigma^2(\text{diag}(W^\top W) + \sigma^2 I)^{-1}$. The "balanced" weights in this case is $V = M^{-1}W^\top$, $M = W^\top W + \sigma^2 I$

Assume optimal $D = D^*$ and balanced weights, we can rewrite the gradients. First, look at the gradient for $V$,

$$\nabla V = \frac{n}{\sigma^2}(W^\top - (W^\top W + \sigma^2 I)V)\Sigma$$
$$= \frac{n}{\sigma^2}((W^\top W + \sigma^2 I)V - (W^\top W + \sigma^2 I)V)\Sigma$$
$$= 0$$

The gradient for $V$ simply forces $V$ to be "balanced" with $W$. Then for $W$,

$$\nabla W = \frac{n}{\sigma^2}(\Sigma V^\top - DW - WV\Sigma V^\top)$$
$$= \frac{n}{\sigma^2}(\Sigma V^\top - \sigma^2(\text{diag}(W^\top W) + \sigma^2 I)^{-1}W - WV\Sigma V^\top)$$
$$= \frac{1}{\sigma^2}(XX^\top V^\top - n\sigma^2\text{diag}(M)^{-1}W - WVXX^\top V^\top)$$
$$= \frac{1}{\sigma^2}(XY^\top - n\sigma^2\text{diag}(M)^{-1}W - WYY^\top)$$
$$= \frac{1}{\sigma^2}(X - WY)Y^\top - n \cdot \text{diag}(M)^{-1}W$$

This is exactly non-uniform $\ell_2$ regularization on $W$. The $\ell_2$ weights are dependent on $W$.

$$\mathrm{diag}(M)^{-1} = \mathrm{diag}(W^\top W + \sigma^2 I)^{-1}$$

# G Connection between the rotation augmented gradient and the Brockett cost function

In this section, we discuss the connection between our rotation augmented gradient and the gradient of the Brockett cost function. In particular, we show that the two updates share similar forms.

Since the Brockett cost function is defined on the Stiefel manifold, we assume throughout this section that $W_1 = W_2^\top$, and $W_2^\top W_2 = I$. Let $\Sigma = \frac{1}{n} X X^\top$ be the data covariance, the Brockett cost function is,

$$\mathrm{Tr}(W_2^\top \Sigma W_2 N) \quad \text{subj. to} \quad W_2^\top W_2 = I_k \ \ (\text{i.e. } W_2 \in \mathrm{St}(k,m))$$

Where $N = \mathrm{diag}(\mu_1,\ldots,\mu_k)$, and $0 < \mu_1 < \cdots < \mu_k$ are constant coefficients. To make the gradient form more consistent with the rotation augmented gradient, we switch the sign of the loss, and reverse the ordering of the diagonal matrix $N$. This does not change the optimization problem, due to the constraint that $W_2$ is on the Stiefel manifold. We define,

$$\mathcal{L}_B(W_2) = -\mathrm{Tr}(W_2^\top \Sigma W_2 D) \quad \text{subj. to} \quad W_2^\top W_2 = I_k$$

Where $D = \mathrm{diag}(d_1,\ldots,d_k)$, $d_1 > \cdots > d_k > 0$. Let $\mathrm{skew}(M) = \frac{1}{2}(M - M^\top)$, the gradient of the cost function on the Stiefel manifold is,

$$\nabla_{W_2}\mathcal{L}_B = -2(I - W_2 W_2^\top)\Sigma W_2 D - W_2 \mathrm{skew}(2W_2^\top \Sigma W_2 D)$$

The gradient descent update in the continuous time limit is,

$$\dot{W}_2 = 2(I - W_2 W_2^\top)\Sigma W_2 D + 2W_2 \mathrm{skew}(W_2^\top \Sigma W_2 D) \tag{51}$$

**Rotation augmented gradient**  With $W_1^\top = W_2$, the rotation augmented gradient update is,

$$\dot{W}_2 = 2(I - W_2 W_2^\top)\Sigma W_2 - 2W_2 \mathrm{skew}(\triangledown(W_2^\top \Sigma W_2)) \tag{52}$$

The updates (51) and (52) appear to have similar forms. We can make the connection more obvious with further manipulation. We express the second term in (51) with the triangular masking operations $\triangledown$ and $\vartriangle$,

$$\begin{aligned}
\mathrm{skew}(W_2^\top \Sigma W_2 D) &= \mathrm{skew}(\triangledown(W_2^\top \Sigma W_2 D) + \vartriangle(W_2^\top \Sigma W_2 D)) \\
&= \mathrm{skew}(\triangledown(W_2^\top \Sigma W_2 D) - \vartriangle(W_2^\top \Sigma W_2 D)^\top) \\
&= \mathrm{skew}(\triangledown(W_2^\top \Sigma W_2)D - \left(\vartriangle(W_2^\top \Sigma W_2)D\right)^\top) \\
&= \mathrm{skew}(\triangledown(W_2^\top \Sigma W_2)D - D\vartriangle(W_2^\top \Sigma W_2)^\top) \\
&= \mathrm{skew}(\triangledown(W_2^\top \Sigma W_2)D - D\triangledown(W_2^\top \Sigma W_2))
\end{aligned}$$

Then, we write the masks explicitly with element-wise multiplications,

$$\begin{aligned}
\mathrm{skew}(W_2^\top \Sigma W_2 D) = \mathrm{skew}(&\left(\begin{bmatrix} 1 & \cdots & 1 \\ & \ddots & \vdots \\ & & 1 \end{bmatrix} \circ W_2^\top \Sigma W_2\right)\begin{bmatrix} d_1 & & \\ & \ddots & \\ & & d_k \end{bmatrix} \\
&- \begin{bmatrix} d_1 & & \\ & \ddots & \\ & & d_k \end{bmatrix}\left(\begin{bmatrix} 1 & \cdots & 1 \\ & \ddots & \vdots \\ & & 1 \end{bmatrix} \circ W_2^\top \Sigma W_2\right)) \\
= \mathrm{skew}\Bigg(&\begin{bmatrix} 0 & d_2 - d_1 & d_3 - d_1 & \cdots & d_k - d_1 \\ & 0 & d_3 - d_2 & \cdots & d_k - d_2 \\ & & \ddots & & \vdots \\ & & & 0 & d_k - d_{k-1} \\ & & & & 0 \end{bmatrix} \circ W_2^\top \Sigma W_2\Bigg)
\end{aligned}$$

Finally, we compare the two updates below,

**Brockett update**

$$\dot{W}_2 = 2(I - W_2 W_2^\top)\Sigma W_2 D - 2W_2 \text{skew}\left(\begin{bmatrix} 0 & d_1 - d_2 & d_1 - d_3 & \cdots & d_1 - d_k \\ & 0 & d_2 - d_3 & \cdots & d_2 - d_k \\ & & \ddots & & \vdots \\ & & & 0 & d_{k-1} - d_k \\ & & & & 0 \end{bmatrix} \circ W_2^\top \Sigma W_2\right)$$

**Rotation augmented gradient update**

$$\dot{W}_2 = 2(I - W_2 W_2^\top)\Sigma W_2 - 2W_2 \text{skew}\left(\begin{bmatrix} 0 & 1 & \cdots & 1 \\ & 0 & \ddots & \vdots \\ & & \ddots & 1 \\ & & & 0 \end{bmatrix} \circ (W_2^\top \Sigma W_2)\right)$$

Both algorithms account for the rotation using the off-diagonal part of $W_2^\top \Sigma W_2$. The rotation augmented gradient applies binary masking, whereas the Brockett update introduces additional coefficients $(d_1, \ldots, d_k)$ that "weights" the rotation.

# H  Experiment details

We provide the experiment details in this section. The code is provided at `https://github.com/XuchanBao/linear-ae`.

## H.1  Convergence to optimal representation

In this section, we give the details of experiments for convergence to the optimal representation on the MNIST dataset (Figure 2 and 3).

The dataset is the MNIST training set, consisting of 60,000 images of size $28 \times 28$ ($m = 784$)). The latent dimension is $k = 20$. The data is pixel-wise centered around zero. Training is done in full-batch mode.

The regularization parameters $\lambda_1, \ldots, \lambda_k$ for the non-uniform $\ell_2$ regularization are chosen to be $\sqrt{\lambda_1} = 0.1$, $\sqrt{\lambda_k} = 0.9$, and $\sqrt{\lambda_2}, \ldots, \sqrt{\lambda_{k-1}}$ equally spaced in between.

The prior probabilities for the nested dropout and the deterministic variant of nested dropout are both chosen to be: $p_B(b) = \rho^b(1 - \rho)$ for $b < k$, and $p_B(k) = 1 - \sum_{b=1}^{k-1} p_B(b)$. We choose $\rho = 0.9$ for our experiments. This is consistent with the geometric distribution recommended in Rippel et al. [24], due to its memoryless property.

The network weights are initialized independently with $\mathcal{N}(0, 10^{-4})$. We experiment with two optimizers: Nesterov accelerated gradient descent with momentum 0.9, and Adam optimizer. The learning rate for each model and each optimizer is searched to be optimal. See Table 2 for the search details, and the optimal learning rates.

## H.2  Scalability to latent representation sizes

The details of the experiments for scalability to latent representation sizes correspond to Figure 4.

The synthetic dataset has 5000 randomly generated data points, each with dimension $m = 1000$. The singular values of the data are equally spaced between 1 and 100. In order to test the scalability of different models to the latent representation sizes, we run experiments with 10 different latent dimension sizes: $k = 2, 5, 10, 20, 50, 100, 200, 300, 400, 500$.

The regularization parameters $\lambda_1, \ldots, \lambda_k$ for the non-uniform $\ell_2$ regularization are chosen to be $\sqrt{\lambda_1} = 0.1$, $\sqrt{\lambda_k} = 10$, and $\sqrt{\lambda_2}, \ldots, \sqrt{\lambda_{k-1}}$ equally spaced in between.

The prior probabilities for the nested dropout and the deterministic variant of nested dropout, the initialization scheme for the network weights, and the optimizers are chosen in the same way as in Section H.1.

We perform a search to find the optimal learning rates for each model, each optimizer with different latent dimensions. See Table 3 for the search details, and Table 4 for the learning rates used in the experiments.

Table 2: Learning rate search values for experiments on MNIST (Figure 2 and 3). The optimal learning rates are labelled in boldface. Note that the Adam optimizer does not apply to RAG.

| Model | Nesterov learning rates | Adam learning rates |
|---|---|---|
| Uniform $\ell_2$ | **1e−3** | **1e−3** |
| Non-uniform $\ell_2$ | 1e−4, **3e−4**, 1e−3, 3e−3 | 1e−3, 3e−3, **1e−2**, 3e−2 |
| Rotation | 1e−3, **3e−3**, 1e−2 | — |
| Nested dropout (nd) | 1e−2, **3e−2**, 1e−1 | 3e−3, **1e−2**, 3e−2, 1e−1 |
| Deterministic nd | 1e−2, **3e−2**, 1e−1 | 3e−3, **1e−2**, 3e−2, 1e−1 |
| Linear VAE | 3e−4, **1e−3**, 3e−3 | 3e−4, **1e−3**, 3e−3 |

Table 3: Learning rate search values for experiments on the synthetic dataset (Figure 4). The optimal learning rates are labelled in boldface. Note that Adam optimizer does not apply to RAG, even though the experiments are shown here.

(a) $k = 20$

| Model | Nesterov learning rates | Adam learning rates |
|---|---|---|
| Non-uniform $\ell_2$ | 1e−4, 3e−4, **1e−3**, 3e−3 | 1e−3, **3e−3**, 1e−2, 3e−2 |
| Rotation | 3e−5, **1e−4**, 3e−4, 1e−3 | 1e−4, **3e−4**, 1e−3 |
| Nested dropout (nd) | 1e−4, 3e−4, **1e−3**, 3e−3 | 1e−3, 3e−3, **1e−2**, 3e−2 |
| Deterministic nd | 1e−4, 3e−4, **1e−3**, 3e−3 | 1e−3, **3e−3**, 1e−2, 3e−2 |
| Linear VAE | 3e−5, 1e−4, **3e−4**, 1e−3 | 3e−4, 1e−3, **3e−3**, 1e−2 |

(b) $k = 200$

| Model | Nesterov learning rates | Adam learning rates |
|---|---|---|
| Non-uniform $\ell_2$ | 1e−4, 3e−4, **1e−3**, 3e−3 | 1e−3, **3e−3**, 1e−2, 3e−2 |
| Rotation | 3e−5, **1e−4**, 3e−4, 1e−3 | 1e−4, **3e−4**, 1e−3 |
| Nested dropout (nd) | 1e−4, 3e−4, **1e−3**, 3e−3 | 3e−4, **1e−3**, **3e−3**, 1e−2 |
| Deterministic nd | 1e−4, 3e−4, **1e−3**, 3e−3 | 1e−3, 3e−3, **1e−2**, 3e−2 |
| Linear VAE | 3e−5, 1e−4, **3e−4**, 1e−3 | 3e−4, **1e−3**, 3e−3, 1e−2 |

(c) $k = 500$

| Model | Adam learning rates |
|---|---|
| Deterministic nd | 3e−3, **1e−2**, 3e−2, 1e−1 |

# I   Additional experiments

## I.1   Non-uniform $\ell_2$ regularization with optimal penalty weights (at global minima)

In this section, we show the experimental results of the learning dynamics of the non-uniform $\ell_2$ regularization on MNIST, with "optimally" chosen $\ell_2$ penalty weights. Specifically, we set the latent dimension $k = 20$, and obtain the $\lambda_1, \ldots, \lambda_k$ values by solving the $\min \max$ optimization problem in (16). These choices of the $\ell_2$ penalty weights are optimal at global minima, because the condition number of the Hessian of the objective at global minima is minimized.

Table 4: Learning rate used for experiments on the synthetic dataset (Figure 4). Note that Adam optimizer does not apply to RAG, even though the experiments are shown here.

(a) Nesterov accelerated gradient descent ($k \leq 50$)

| $k$ | 2 | 5 | 10 | 20 | 50 |
|---|---|---|---|---|---|
| Non-uniform $\ell_2$ | 1e−3 | 1e−3 | 1e−3 | 1e−3 | 1e−3 |
| Rotation | 1e−4 | 1e−4 | 1e−4 | 1e−4 | 1e−4 |
| Nested dropout (nd) | 1e−3 | 1e−3 | 1e−3 | 1e−3 | 1e−3 |
| Deterministic nd | 1e−3 | 1e−3 | 1e−3 | 1e−3 | 1e−3 |
| Linear VAE | 3e−4 | 3e−4 | 3e−4 | 3e−4 | 3e−4 |

(b) Nesterov accelerated gradient descent ($k \geq 100$)

| $k$ | 100 | 200 | 300 | 400 | 500 |
|---|---|---|---|---|---|
| Non-uniform $\ell_2$ | 1e−3 | 1e−3 | 1e−3 | 1e−3 | 1e−3 |
| Rotation | 1e−4 | 1e−4 | 1e−4 | 1e−4 | 1e−4 |
| Nested dropout (nd) | 1e−3 | 1e−3 | 1e−3 | 1e−3 | 1e−3 |
| Deterministic nd | 1e−3 | 1e−3 | 1e−3 | 1e−3 | 1e−3 |
| Linear VAE | 3e−4 | 3e−4 | 3e−4 | 3e−4 | 3e−4 |

(c) Adam optimizer ($k \leq 50$)

| $k$ | 2 | 5 | 10 | 20 | 50 |
|---|---|---|---|---|---|
| Non-uniform $\ell_2$ | 3e−3 | 3e−3 | 3e−3 | 3e−3 | 3e−3 |
| Rotation | 3e−4 | 3e−4 | 3e−4 | 3e−4 | 3e−4 |
| Nested dropout (nd) | 1e−2 | 1e−2 | 1e−2 | 1e−2 | 3e−3 |
| Deterministic nd | 3e−3 | 3e−3 | 3e−3 | 3e−3 | 3e−3 |
| Linear VAE | 3e−3 | 3e−3 | 3e−3 | 3e−3 | 1e−3 |

(d) Adam optimizer ($k \geq 100$)

| $k$ | 100 | 200 | 300 | 400 | 500 |
|---|---|---|---|---|---|
| Non-uniform $\ell_2$ | 3e−3 | 3e−3 | 3e−3 | 3e−3 | 3e−3 |
| Rotation | 3e−4 | 3e−4 | 3e−4 | 3e−4 | 3e−4 |
| Nested dropout (nd) | 3e−3 | 3e−3 | 3e−3 | 3e−3 | 3e−3 |
| Deterministic nd | 1e−2 | 1e−2 | 1e−2 | 1e−2 | 1e−2 |
| Linear VAE | 1e−3 | 1e−3 | 1e−3 | 1e−3 | 1e−3 |

In practice, the $\ell_2$ penalty weights in Figure 5 are not accessible without knowing the $\sigma$ values of the dataset. However, we show in Figure 6 that even with this knowledge, using the $\lambda$ values optimal at global optima significantly slows down the initial phase of training. This means that these $\lambda$ values are suboptimal away from global optima. In general, it is difficult to determine the $\lambda$ values that are optimal for the overall training process. This contributes to the weakness of symmetry breaking by the non-uniform $\ell_2$ regularization.

## I.2 Mini-batch training on MNIST

In this section, we show the learning dynamics of the models in Section 7 trained on MNIST using mini-batches. The uniform $\ell_2$ regularized LAE is not included, as it doesn't recover the axis-aligned solutions. Figure 7 and 8 show the learning dynamics with $k = 20$ and mini-batch size 1000 and 100, respectively. We observe similar results as in the full-batch setting (Figure 2), with additional stochasticity introduced by mini-batch training.

Figure 5: Optimal $\ell_2$ penalty weights on MNIST, with $k = 20$.

(a) Axis-alignment

(b) Subspace convergence

Figure 6: Learning dynamics of non-uniform $\ell_2$ regularized LAEs on the MNIST ($k = 20$), with different choices of penalty weight values. All models are trained with Adam optimizer for 1000 epochs. The optimal $\lambda$ values are as in Figure 5. Results with different learning rates are shown, provided that the learning rates are small enough to maintain training stability.

(a) Axis-alignment

(b) Subspace convergence

Figure 7: Learning dynamics of different LAE / linear VAE models trained on MNIST ($k = 20$), with mini-batch size 1000. Solid lines represent models trained using gradient descent with Nesterov momentum 0.9. Dashed lines represent models trained with Adam optimizer.

(a) Axis-alignment

(b) Subspace convergence

Figure 8: Learning dynamics of different LAE / linear VAE models trained on MNIST ($k = 20$), with mini-batch size 100. Solid lines represent models trained using gradient descent with Nesterov momentum 0.9. Dashed lines represent models trained with Adam optimizer.

## Footnotes

[4]This result is a simple consequence of properties of positive semi-definite matrices