[Reviews · NeurIPS 2020]

Review 1

Summary and Contributions: The presented

Strengths: The paper is clearly structured and relatively easy to follow. The analysis of the loss landscape for the non-uniformly l2-regularized linear autoencoder appears thorough and sound (I have only superficially checked the proofs in the appendix for plausibility). The idea of using non-uniform l2 regularization for obtaining ordered representations is intriguing and could prove useful in other applications of neural networks. Additionally, the paper presents a deterministic variant of the regularization term given by nested dropout. The introduced modification to the gradient-based update rule could improve convergence of neural networks, where the problem of weakly-broken rotational symmetry exists.

Weaknesses: My main concern with this submission lies in its utility for the community. I do not see how the presented findings can be generalized to any neural network used in practice. On the other hand, I acknowledge that the main benefit of research is not in the immediate application of its findings. Nevertheless, I would appreciate a discussion of this aspect. For example, the utility of the proposed update rule should be discussed more in-depth. From what I understand it is only useful for problems withouth mini-batching and where there is a subspace with weakly

Correctness: The derivations and proofs are correct, as far as I can tell.

Clarity: The paper has a clear and intuitive structure, which makes it easy to follow its logic. The writing is somewhat dry, as can be expected of a mathematical paper.

Relation to Prior Work: The relevant work I know of is covered with the exception of papers investigating the link between L2 regularization and dropout (e.g. Wager et al. "Dropout Training as Adaptive Regularization"), which I believe to be relevant here.

Reproducibility: Yes

Additional Feedback:


Review 2

Summary and Contributions: I have read the rebuttal. Thank you **** The authors study symmetry breaking properties of two regularization schemes for linear autoencodes, and provide a new algorithm to circumvent the slow convergence stemming out of illconditioning introduced by such a scheme.

Strengths: The paper is very well written and delineates its contributions well. The paper makes an important contribution to the community – discussing the symmetry breaking properties of certain regularization schemes, closely studying convergence properties based on loss landscape and designing algorithms to circumvent the problematic issues that slow down the convergence. The authors build upon the previous work of Kunin et al who showed a slightly weaker form of symmetry breaking (still retains orthogonality) using uniform l2 regularization and show that non-uniform regularization breaks orthogonality and only retains reflection. This is intuitive, since penalizing different directions differently does not allow for orthogonality anymore. While the proof methodologies are not new, the proofs themselves are novel and the results interesting. The close analysis of bad conditioning that arises out of the non-uniform regularization and the two-phase split of convergence with the latter slower phase attributed to breaking the rotational symmetry. The authors then modify the gradient descent algorithm to address this issue. This is not straight-forward, as noted by the authors. The noted connections of this proposed algorithm to Hebbian algorithm is also interesting.

Weaknesses: Minor gripes: The links for references did not seem to be working for some reason, please fix as it makes jumping around for references and equations harder. Also, it would be nice if the authors could expand more on future directions. E.g. the development of the augmented gradient method as equivalent to applying a gauge field, this sentence adds zero information without a couple of more sentences explaining the said field, and putting quotes around gauge field does not help at all. The authors have shied away from probabilistic interpretation studied in Kunin et al. It seems the non-uniform regularization should have a straightforward generalization. What about the nested dropout ? Is there a reason to not make these connections in this work?

Correctness: I would think so. I have gone through the main claims/proofs.

Clarity: Definitely yes.

Relation to Prior Work: Yes, the connection to previous works is well-highlighted, including the attribution to results, as well as proof techniques.

Reproducibility: Yes

Additional Feedback:


Review 3

Summary and Contributions: Post-rebuttal update: Thank you for clarifying on the motivation of this work as well as the mistake in proof. I'm updating my score since the two-stage convergence behavior is interesting, and the proposed algorithm has interesting connections to prior work. However, I'm not sure if the modified linear AE model is the best model to understand the slowness of learning NN representations, as the regularization schemes seem somewhat artificial and doesn't seem to correspond to any commonly-used algorithms. From a probabilistic perspective, it's also unclear why we would assign an arbitrary non-uniform prior when we don't have any knowledge about their scales, e.g. is there any gain from choosing a more correct prior about the scales? I think further discussion on these issues would greatly enhance this work. ====== Building upon Kunin et al (2019), this work investigates ways for the regularized linear autoencoder to recover the original principal components (up to inconsequential ambiguities such as permutation). It is shown that both non-uniform L2 regularization and nested dropout lead to such recovery, but ordinary GD using these objectives suffers from slow convergence. An alternative optimization algorithm is developed to accelerate convergence, and is connected to the generalized Hebbian algorithm.

Strengths: This work established new identifiability results, identified the convergence issue of OGD, and proposed an optimization algorithm connected to GHA.

Weaknesses: * Proof of theorem 1 appears incomplete; see below. * Comparing with previous work, the algorithms investigated in this work are less related to those used in practice. While I don't think this issue is deal-breaking, it nonetheless make the scope of this work more limited.

Correctness: I'm not sure if the proof for Theorem 1 is correct: on L593-594, you claim that if \Lambda is a diagonal matrix with positive diagonal elements, v^T \Lambda AA^T v >= 0 will always hold. This is not true; consider AA^T=[[1,-1],[-1,1]], v=[1,0.5], \Lambda v=[1,2].

Clarity: This paper is well written.

Relation to Prior Work: The related work section is mostly thorough, and difference from prior work is clearly discussed. However, * The following concurrent work (according to NeurIPS guidelines) investigates the same identifiability issue, and should probably be discussed in a revised version: Oftadeh et al, Eliminating the Invariance on the Loss Landscape of Linear Autoencoders. ICML 2020. * Regarding connection between linear VAE trained with ELBO and pPCA, you should probably cite the following works, which are earlier than [17]. Dai, Bin, et al. "Connections with robust PCA and the role of emergent sparsity in variational autoencoder models." The Journal of Machine Learning Research 19.1 (2018): 1573-1614. Rolinek, Michal, Dominik Zietlow, and Georg Martius. "Variational autoencoders pursue pca directions (by accident)." Proceedings of the IEEE Conference on Computer Vision and Pattern Recognition. 2019.

Reproducibility: Yes

Additional Feedback: I will update my review once my question regarding correctness is clarified.


Review 4

Summary and Contributions: The authors further theoretical understanding of learning the optimal representations for data using linear autoencoders (LAEs) with different gradient based optimizers. They provide two different regularization methods, non-uniform L2 regularization and deterministic nested dropout, to allow for convergence to the optimal axis-aligned representation. They provide explanations for why such convergence is slow via condition number analysis, and present a modified learning algorithm that greatly enhances the speed of convergence to the optimal axis-aligned solution.

Strengths: The authors further previous work regarding breaking symmetry of the set of optimal solutions for a LAE. The contribution is significant, as they further understanding of why convergence towards the optimal solution might be slow, and they provide an algorithm that shows promising empirical results. The theoretical grounding for the algorithm is good, as they are able to provide a local convergence rate.

Weaknesses: A discussion regarding how the regularization matrix Lambda is chosen for the non-uniform L2 regularized objective would be helpful, since it is no longer just one regularization parameter. There is an assumption made stating that the regularization terms should be smaller than the respective singular values during the characterization of the stationary points (Theorem 3). The authors should explain how the weights are known to satisfy this condition. An updated figure like figure 1 showing the convergence trajectories of the various methods on the loss landscape would be helpful.

Correctness: The claims and method are correct and seem to work empirically. Proofs for all lemmas and theorems are present in the supplementary methods and are sound.

Clarity: The paper is well written and it is easy to follow logically. There are a few minor inconsistencies in the writing. In the experiments section, the text of the work states that convergence in figure 4 is plotted until a value of .25 is reached, but in the figure a value of .3 is stated. It is slightly unclear whether the models that use the augmented gradient algorithm also uses an Adam optimizer, or if the models trained using an Adam optimizer only have a different objective (i.e. non-uniform L2 or deterministic dropout) since the Adam trained models seem to converge the best. 

Relation to Prior Work: A good background discussion is provided on prior work and how this work expands upon prior work. The work is clearly contextualized.

Reproducibility: Yes

Additional Feedback: ** after author rebuttal ** Thank you for the response. I suggest the authors include in the revised text the discussion on choosing lambda matrix in the rebuttal.

[Author Response · NeurIPS 2020]

We thank all reviewers for their helpful reviews. Please see our response below.

**Correctness of Theorem 1 proof** Thank you R3 for pointing out the mistake in the current proof. The mistake is
in the very last step of the proof, where we tried to show $W_1 - W_2^\top = 0$ (lines 593-594). Fortunately, we have the
following fix that asserts the correctness of Theorem 1. We hope R3 will update their score based on this revised proof.

From Lemma 2 (line 583), $C = \frac{1}{n}(I - W_2 W_1) X X^\top \in \mathbb{R}^{m \times m}$ is positive semi-definite, and $\Lambda^2 = \mathrm{diag}(\lambda_1, \ldots, \lambda_k)$
is positive definite. Define $A = W_1 - W_2^\top \in \mathbb{R}^{k \times m}$. We prove below that $A = 0$ follows from line 592 which states

$$\forall v \in \mathbb{R}^k, \quad 0 = v^\top A C A^\top v + v^\top \Lambda^2 A A^\top v \tag{1}$$

*Proof.* Since $A C A^\top \succeq 0$, we have $\forall v, \; v^\top A C A^\top v \geq 0$. Hence, from (1), $\forall v, v^\top \Lambda^2 A A^\top v \leq 0$. Consider setting
$v = e_i$, the $i^{th}$ coordinate vector in $\mathbb{R}^k$ ($i^{th}$ entry is 1, and all others are 0). We must have $e_i^\top \Lambda^2 A A^\top e_i = \lambda_i^2 \|A_i\|_2^2 \leq 0$
($A_i$ denotes the $i^{th}$ row of $A$). Since $\lambda_i > 0$, we have $A_i = 0$. Since this holds for all $i = 1, \ldots, k$, we have $A = 0$. $\square$

**How to choose the non-uniform regularization parameters (R4)** This is a great
question. It's indeed difficult to choose an "optimal" set of $\lambda$ values for the non-
uniform $\ell_2$ regularization (see below for justifications). However, note that our goal
is the analysis, and that the difficulty in choosing the $\lambda$ values a priori contributes
to the argument of weak symmetry breaking by the non-uniform $\ell_2$ regularization.

Figure 1: Optimal $\lambda$ values.

As for why it is difficult to choose an "optimal" set of $\lambda$ values, note that optimal
values at global minima are not optimal in general. At global minima, the optimal $\lambda$
values can be obtained by solving the $\min \max$ optimization in line 509 (an example
of such optimized $\lambda$ values for MNIST, $k = 20$ is shown in Figure 1). However, this
set of $\lambda$ values concentrate on the larger side, and significantly slow down learning
(suboptimal *away from* the global minima). In our experiments, we first make a (rough) estimate of the $k^{th}$ largest data
singular value, and linearly space the $\lambda$ values from 0 to the estimated $\sigma_k$ value (see Appendix G for details).

**Practical utility of the analyzed algorithms (R1, R3)** A big part of our mo-
tivation was to understand the slowness of learning neural net representations,
as opposed to reducing loss — a topic of immense practical importance, as
evidenced by the recent flurry of interest on early training effects. Linear au-
toencoders are one of the few examples where we can determine exactly what
representation *ought* to be learned, which makes them a particularly useful
model system for understanding convergence of representations.

Figure 2: Mini-batch experiment.

**Probabilistic interpretations of the non-uniform $\ell_2$ regularization and the**
**nested dropout?** As pointed out by R2, the probabilistic interpretation of non-
uniform $\ell_2$ regularization is a straightforward generalization of that studied
in Kunin et al. In particular, we can assign a diagonal Gaussian prior to the
weights whose precision parameters equal the $\lambda_i$ value for the corresponding
latent dimension. There is a well-known Bayesian interpretation of dropout
[Gal and Ghahramani, 2016], and extending this to nested dropout is an interesting direction for future work.

**Relation to additional prior work** Compared to Wager et al. [2013] (R1), which discusses the connection of
dropout and adaptive $\ell_2$ regularization in generalized linear models, we work with a different class of models (linear
autoencoders) and study a different type of dropout (nested dropout in the latent units). Nevertheless, it is an interesting
future direction to investigate the connection between the deterministic nested dropout and common types of regularizers.

Concurrent work [Oftadeh et al., 2020] (pointed out by R3) addresses the identifiability issue in linear autoencoders by
proposing a new loss function. The loss function proposed in Oftadeh et al. is a special case of deterministic nested
dropout (section 5 of our paper) where the prior $p_B(\cdot)$ is a uniform distribution. Oftadeh et al. show that the local
minima correspond to ordered, axis-aligned representations, but do not analyze the speed of convergence under the new
loss. Hence, our analysis provides additional insight into their method. We will include the above discussions, as well
as the additional citations regarding the connections between linear VAEs and pPCA (R3) in the revised paper.

**Mini-batch training for the rotation augmented gradient? (R1)** We did additional experiments on MNIST using
the rotation augmented gradient, with various batch sizes and $k = 20$ (Figure 2). The results show that the rotation
augmented gradient works well with mini-batches. Larger batches improve per-epoch convergence up to a point of
diminishing returns, similarly to standard models and algorithms (e.g. Shallue et al., JMLR 2019).

**Writing & clarity** We thank R2 and R4 for the writing & clarity suggestions, and will address them in the revised
paper. Note that the preconditioning of the Adam optimizer is not compatible with the rotation augmented gradient, so
only the SGD is relevant for this algorithm (see Appendix G for more experimental details).

[Meta-Review · NeurIPS 2020]

This paper investigates ways for the regularized linear autoencoder to recover the original principal components of a matrix, and it shows that non-uniform L2 regularization and nested dropout lead to such recovery, ordinary GD using these objectives suffers from slow convergence, a new alternative optimization algorithm can accelerate convergence, and this new algorithm is connected to a Hebbian algorithm. The paper is well written and makes a nice contribution. All reviewers were positive, and several reviewers improved their scores in light of the author responses and subsequent discussion.